Title:
**Variations in airborne bacterial communities at high altitudes over the Noto**
**Peninsula (Japan) in response to Asian dust events**
Authors:
Teruya Maki *[a], Kazutaka Hara[b], Ayumu Iwata[c], Kevin C. Lee[d], Kei Kawai[e], Kenji Kai[e],
Fumihisa Kobayashi[f], Stephen B. Pointing[d], Stephen Archer[d], Hiroshi Hasegawa[a], and
Yasunobu Iwasaka[g]
Author Affiliations:
[a] College of Science and Engineering, Kanazawa University, Kakuma, Kanazawa,
Ishikawa, 920-1192, Japan.
[b] National Institute for Environmental Studies, Tsukuba, Ibaraki 305-8506, Japan.
[c] Graduate school of Natural Science and Technology, Kanazawa University, Kakuma,
Ishikawa, 920-1192, Japan.
[d] School of Applied Sciences, Auckland University of Technology, Private Bag 92006,
Auckland 1142, New Zealand.
[e] Graduate School of Environmental Studies, Nagoya University; Furocho, Chikusaku,
Nagoya, 464-8601, Japan.
[f] Graduate School of Science and Technology, Hirosaki University, Bunkyo-cho 3,
Hirosaki, Aomori, 036-8561, Japan.
[g] Community Research Service Group, University of Shiga Prefecture, 2500
Yasakamachi, Hikoneshi, Shiga, 522-8533, Japan.
*Corresponding author:
Tel: +81-(0) 76-234-4793, Fax: +81-(0) 76-234-4800
E-mail: makiteru@se.kanazawa-u.ac.jp
**Abstract**
Aerosol particles, including airborne microorganisms, are transported through the
free troposphere from the Asian continental area to the downwind area in East Asia and
can influence climate changes, ecosystem dynamics, and human health. However, the
variations present in airborne bacterial communities in the free troposphere over
downwind areas are poorly understood, and there are few studies that provide an
in-depth examination of the effects of long-range transport of aerosols (natural and
anthropogenic particles) on bacterial variations. In this study, the vertical distributions
of airborne bacterial communities at high altitudes were investigated and the bacterial
variations were compared between dust events and non-dust events.
Aerosols were collected at three altitudes from ground level to the free troposphere
(upper level: 3,000 m or 2,500 m; middle level: 1,200 m or 500 m; and low level: 10 m)
during Asian dust events and non-dust events over the Noto Peninsula, Japan, where
westerly winds carry aerosols from the Asian continental areas. During Asian dust
events, air masses at high altitudes were transported from the Asian continental area by
westerly winds, and Laser Imaging Detection and Ranging (LIDAR) data indicated high
concentrations of non-spherical particles, suggesting that dust-sand particles were
transported from the central desert regions of Asia. The air samples collected during the
dust events contained 10–100 times higher concentrations of microscopic fluorescent
particles and Optical Particle Counter (OPC) measured particles than in non-dust events.
The air masses of non-dust events contained lower amounts of dust-sand particles.
Additionally, some air samples showed relatively high levels of black carbon, which
were likely transported from the Asian continental coasts. Moreover, during the dust
events, microbial particles at altitudes of >1,200 m increased to the concentrations
ranging from 1.2 x $10^6$ particles $m^{-3}$ to 6.6 x $10^6$ particles $m^{-3}$. In contrast, when dust
events disappeared, the microbial particles at >1,200 m decreased slightly to
microbial-particle concentrations ranging from 6.4 x $10^4$ particles $m^{-3}$ to 8.9 x $10^5$
particles $m^{-3}$.
High-throughput sequencing technology targeting 16S rRNA genes (16S rDNA)
revealed that the bacterial communities collected at high altitudes (from 500 m to 3,000
m) during dust events exhibited higher diversities and were predominantly composed of
natural-sand/terrestrial bacteria, such as *Bacillus* members. During non-dust periods,
airborne bacteria at high altitudes were mainly composed of anthropogenic/terrestrial
bacteria (Actinobacteria), marine bacteria (Cyanobacteria and Alphaproteobacteria), and
plant-associated bacteria (Gammaproteobacteria), which shifted in composition in
correspondence with the origins of the air masses and the meteorological conditions.
The airborne bacterial structures at high altitudes suggested remarkable changes in
response to air mass sources, which contributed to the increases in community richness
and to the domination of a few bacterial taxa.


## 1. Introduction

Airborne microorganisms (bioaerosols) associated with desert-sand and anthropogenic particles were transported through free troposphere from the Asian continents to downwind regions of East Asia and can influence climate changes, ecosystem dynamics, and human health (Iwasaka et al., 2009). Natural dust events from the Asian desert regions carry airborne microorganisms, supporting atmospheric microbial dispersals (Griffin et al., 2007; Maki et al., 2010; Pointing and Belnap, 2014). Haze days caused by anthropogenic particles from Asian continents also affect airborne microbial abundance and endotoxin levels (Wei et al., 2016). Some studies demonstrated that Asian dust events, including natural and anthropogenic particles, cause vertical mixture of bioaerosols in downwind areas, such as in Japan (Huang et al., 2015b; Sugimoto et al., 2012; Maki et al., 2015).

Bioaerosols, which include bacteria, fungi, and viruses, are transported from ground environments to the free troposphere and account for a substantial proportion of organic aerosols (Jaenicke, 2005). Bioaerosols are thought to influence atmospheric processes by participating in atmospheric chemical reactions and in the formation of cloud-nucleating particles (Pratt et al., 2009; Morris et al., 2011; Hara et al., 2016b). Indeed, airborne microorganisms act as ice nuclei that are related to ice-cloud formation processes (Möhler et al., 2007; Delort et al., 2010; Creamean et al., 2013; Joly et al., 2013). In particular, ice-nucleation activating proteins of some microorganims, such as *Pseudomonas syringae*, *Xanthomonas campestris* and *Erwinia herbicola*, exhibit high nucleation activities, initiating ice formation at relatively warm temperatures (greater than -5 °C) (Morris et al. 2004) in comparison to the inorganic ice-nucleating particles,

such as potassium feldspar (approximately -8 °C) (Atkinson et al. 2013). Ice-nucleating
particles that originate from bioaerosols are believed to activate ice formation more
efficiently than inorganic substances (Hoose and Möhler, 2012; Murray et al. 2012), and
are primary contributors of rapid ice-cloud formation even at low concentrations in the
clouds at temperatures between -8 °C and -3 °C (Hallett and Mossop, 1974).
Bioaerosols are key factors for elucidating the detailed mechanisms of ice-cloud
formation and precipitation over East Asia (Hara et al., 2016ab), but the microbial
characteristics of bioaerosols transported over long distances by Asian-dust events are
still unclear. Furthermore, the microorganisms transported by Asian dust events increase
the allergenic burden, consequently inducing asthma incidences (Ichinose et al., 2005)
and contributing to the dispersal of diseases such as Kawasaki disease (Rodó et al.,
2011) and rust diseases (Brown and Hovmøller, 2002).
In downwind areas of East Asia, the atmospheric bacterial dynamics at high
altitudes should be investigated in order to understand the ecological and meteorological
influences of airborne bacteria as well as their long-range dispersion. Meteorological
shifts and dust events can dramatically alter airborne bacterial communities at high
altitudes in Japan (Maki et al., 2013 and 2015) because of air masses that originate from
heterogeneous environments, including marine, mountainous, urban, and desert areas.
The airborne microorganisms around North American mountains (2,700 m above sea
level) were also found to increase their species diversities in response to Asian dust
events (Smith et al., 2013). High-throughput sequencing technology can generate large
numbers of nucleotide sequences and the sequencing database has played an important
role for investigation of airborne bacterial compositions (Brodie et al., 2007; Woo et al.,
2013). Indeed, the analyses using high-throughput sequencing has demonstrated that
airborne bacterial populations at ground levels change in response to pollutants from
Beijing (Cao et al., 2014) and African dust events (Mazar et al., 2016). To investigate
their long-range transported bacteria while avoiding the ground-surface contaminations,
the bioaerosol samples collected at high altitudes by aircrafts were analyzed using
high-throughput sequencing, showing the airborne microbial diversities at high altitudes,
ranging from 1,000 m to 3,000 m (DeLeon-Rodriguez et al., 2013; Maki et al., 2015).
There are also a few studies on the vertical bacterial distribution from the ground level
to the troposphere (DeLeon-Rodriguez et al., 2013; Maki et al., 2015). Nonetheless,
while some variations were observed, the specific changes in tropospheric bioaerosols
over East Asia, and, in particular, differences between Asian dust and non-dust events
remain poorly understood.
Organic aerosol particles, such as bioaerosols, account for high rates of
tropospheric aerosols, ranging from 30 % to 80 % (Jaenicke, 2005), and fluctuate at
high concentrations, ranging from $10^3$ to $10^5$ particles m$^{-3}$, under the boundary layer at
4,000 m above the ground (Twohy et al., 2016). Epifluorescence microscopy using
fluorescent-dye staining is a useful tool for observation and determination of microbial
particles in the atmosphere, demonstrating that the biomass of airborne microorganisms
increased 10– to 100–fold during Asian-dust events (Hara et al., 2012, Maki et al.,
2014). Under a fluorescence microscope, DNA in microbial particles fluoresce blue
when stained with 4, 6-diamidino-2-phenylindole (DAPI) (Russell et al., 1974), and
organic materials aggregated with proteins and microbial cell components were
confirmed as yellow fluorescence particles (Mostajir et al., 1995). Mineral particles
(white particles) and black carbon (black particles) can also be observed as background
fluorescence in microscopic observation fields (Maki et al., 2014). Accordingly, several
DAPI-stained particles could be detected in air samples collected from all over Japan
during dust events (Maki et al., 2013) and can be used as indicators for evaluating the
amounts of some aerosol species during dust events.
In this study, the bacterial communities from different altitudes around the
Japanese islands were compared to identify the potential influences of long-range
transported air masses on tropospheric bacteria. We used a helicopter for collecting air
samples at altitudes ranging from 1,200 m to 3,000 m over the Noto Peninsula, Japan.
Helicopter sampling was used to collect chemical components at high altitudes, which
has previously been used to avoid contamination from the downwash created by
spinning rotors (Watanabe et al., 2016). This air sampling method can directly collect
aerosols moving from Asian continents or marine areas to Japan. We estimated the air
mass conditions using the meteorological data obtained during the sampling periods,
and determined aerosol amounts by using meteorological monitoring and
epifluorescence microscopic observation. Bacterial community structures were analyzed
by using high-throughput sequencing targeting bacterial 16S rRNA genes (16S rDNA).

**2. Experiments**

*2.1. Sampling*
Aerosol sampling using a helicopter (R44; Robinson, CA, USA) was performed
over coastal areas from Uchinada (36º67N, 136º64E) to Hakui (36º92N, 136º76E) in the
Noto Peninsula, Japan. Both cities are located on the western coast of the Noto
Peninsula where aerosols arrive from continental areas across the Sea of Japan and are
mixed with local aerosols (Fig. 1). The helicopter traveled 20 km northwest from
Kanazawa to Uchinada; air sampling was continuously conducted from Uchinada to the
northern coastal areas. To compare the vertical distributions of airborne bacteria during
dust and non-dust events, air samples were collected using a helicopter at the 1 to 3
altitudes ranging from 500 m to 3,000 m above ground level (Table 1). Air samples
from low altitude regions (10 m above ground level) were collected from the roof of a
building located at Taki bay in Hakui (36º92 N, 136º76 E). To compare the vertical
bacterial distribution, aerosol samples were collected during the daytime (from 9:00
Japanese standard time [JST; UTC + 9 h] to 16:30 JST) on March 19, 2013; April 28,
2013; March 28, 2014; and March 20, 2015. These samples were collected at the
following altitude sets; (1) 2,500 m, 1,200 m, and 10 m; (2) 3,000 m, 1,200 m, and 10
m; (3) 3,000 m, 1,200 m, and 10 m; and (4) 2,500 m and 500 m, respectively, and
samples were labeled as shown in Table 1. To investigate the bacterial changes at
altitudes in response to time, temporal transect at the altitude of 1,200 m was prepared
for seven days – the 23rd, 24th, 25th, and 29th of March 2014 and the 16th, 17th, and
21st of March 2015 – and the sample names are showed in Table 1.
Air samples were collected through sterilized polycarbonate filters (0.22-μm pore
size; Whatman, Tokyo, Japan) with sterilized filter holders (Swinnex Filter holder;
Merck, Darmstadt, German) connected to an air pump. At the sterilization processes, the
filters and the filter-holder parts were irradiated separately under UV light for 1.0 h and
the filter holders attached with the filters were autoclaved at 121 ℃ for 20 min. Air
sampling was performed with a flow rates of 5 L min$^{-1}$ over sampling periods from 0.2
h to 1.0 h. Triplicate sampling filters were obtained for each altitude. During helicopter
sampling, outside air was transferred from a window to the bioaerosols-sampling inlet,
which was sterilized by autoclaving and UV irradiation. The sterilized filter holders
were inserted into the sampling inlet to avoid contamination. To collect air particles at
an altitude of 10 m, we used filter holders fixed on a 3 m stick, which was placed on the
roof of a building (Maki et al., 2014).
In total, 18 air samples were obtained during the sampling periods (Table 1). Of
the two filters used to collect each sample, one filter was used to determine the
particulate abundances under fluorescence microscopy, and the other was stored at
-80°C before the extraction of genomic DNA for analysis of bacterial compositions.

*2.2. Characteristics and trajectories of air masses*
Information regarding weather conditions (temperature, relative humidity, and
pressure) was gathered. During the helicopter flight, outside air was transferred from a
window into the meteorological-measurement inlet, into which the adaptor of the
measurement device (TR-73U; T&D Corporation, Matsumoto, Japan) was inserted, and
the temperature, relative humidity, and pressures were sequentially measured. The
temperature and relative humidity at an altitude of 10 m were also measured on the roof
of a building in Hakui. The depolarization ratio, which was measured by Laser Imaging
Detection and Ranging (LIDAR) measurements at Toyama, has been used for the
detection of non-spherical aerosols, such as mineral dust particles and/or sea salts.
To track the transport pathways of air masses, 72 h back trajectories were
calculated using the National Oceanic and Atmospheric Administration (NOAA)
HYbrid Single Particle Lagrangian Integrated Trajectory (HYSPLIT) model
(http://www.arl.noaa.gov/HYSPLIT.php). The coordinator of Hakui was used as the
back trajectory starting point at several altitudes from 10 m to 3,000 m above ground
level to estimate the trajectories of the air masses.

*2.3. Determination of particle abundance*
The air particles at each altitude were measured using an optical particle counter
(OPC: Rion, Tokyo, Japan). The OPC device was connected to the
meteorological-measurement inlet. The air particles at an altitude of 10 m were also
counted using the OPC device placed on the roof of a building.
Fluorescent particles stained with DAPI were also counted via epifluorescence
microscopy. Within 2 h of sampling, 1 mL of 1 % paraformaldehyde was added to one
of the filters to fix the aerosols. After a 1 h incubation, the filter was stained with DAPI
at a final concentration of 0.5 µg mL$^{-1}$ for 15 min (Russell et al., 1974). Next, the filter
was placed on a slide in a drop of low-fluorescence immersion oil (Type-F
IMMOIL-F30CC, Olympus, Tokyo, Japan). A second drop of oil was added, and a
coverslip was placed on top. Particles on the filter were observed using a fluorescence
microscope (BX-51, Olympus, Tokyo, Japan) with a UV excitation system. A filter
transect was scanned, and the four categorized particles, including white fluorescent
particles, blue fluorescent particles (microbial particles), yellow fluorescent particles,
and black particles, on the filter transect were counted using a previously reported
observational technique (Maki et al., 2014). The TA connections in DNA sequences of
microbial particles are bound with DAPI, emitting clear blue fluorescence. However,
the aggregation of organic matter might also accumulate DAPI at high amounts emitting
yellow fluorescence, which is due to formation of a compound with DAPI. Mineral
particles often have white autofluorescence or emit weak-blue (mostly white)
fluorescence originating from residues of DAPI on the particle surfaces and can be
identified on the weak blight background of microscopic observation fields. The black
color of black carbon can be identified in the background. The detection limit of aerosol
particle concentration was $1.1 \times 10^4$ particles m$^{-3}$ of air.

*2.4. Analysis of bacterial community structures using MiSeq sequencing analysis*
*targeting 16S rDNA sequences*

After the aerosol particles on the other two filters were suspended in 3 mL of

sterile 0.6 % NaCl solution, the particles were pelleted by centrifugation at $20,000 \times g$
for 10 min. The genomic DNA (gDNA) was then extracted from the particle pellets
using sodium dodecyl sulfate, proteinase K, and lysozyme and purified by
phenol-chloroform extraction as previously described (Maki et al., 2008). The bacterial
community structure was determined using MiSeq DNA sequencing, which facilitates
multiplexed partial sequencing of 16S rDNA. Fragments of 16S rDNA (approximately
500 bp) were amplified from the extracted gDNA by PCR using the universal 16S
rDNA bacterial primers 515F (5′- Seq A -TGTGCCAGCMGCCGCGGTAA-3′) and
806R (5′- Seq B -GGACTACHVGGGTWTCTAAT-3′) (Caporaso et al., 2011), where
Seq A and Seq B represent the nucleotide sequences bounded by the second set of PCR
primers described below. The PCR amplicon sequences covered the variable region V4
of the 16S rRNA gene. Thermal cycling was performed using a thermocycler (Program
Temp Control System PC-700; ASTEC, Fukuoka, Japan) under the following
conditions: denaturation at 94°C for 1 min, annealing at 52°C for 2 min, and extension
at 72°C for 2 min for 20 cycles. Fragments of 16S rDNA in PCR products were
amplified again using the second PCR forward primer (5′- Adaptor C - xxxxxxxx - Seq
A -3′) and reverse primer (5′- Adaptor D - Seq B -3′), where Adaptors C and D were
used for the Miseq sequencing reaction. The sequences "xxxxxxxx" comprise an 8
nucleotide sequence tag designed for sample identification barcoding. Thermal cycling
was performed under the following conditions: denaturation at 94°C for 1 min,
annealing at 59°C for 2 min, and extension at 72°C for 2 min for 15 cycles. PCR
amplicons were purified using the MonoFas DNA purification kit (GL Sciences, Tokyo,
Japan). PCR amplicons from each sample were pooled at approximately equal amounts
into a single sequencing tube on a MiSeq Genome Sequencer (Illumina, CA, USA)
machine. The sequences obtained for each sample were demultiplexed based on the tag,
including the 8 nucleotide sequence. After removal of the tags, an average read length
of 450 bp was obtained. Negative controls (no template and extraction products from
unused filters) were prepared in the DNA extraction process to check for contamination.
The amount of gDNA extracted from air samples ranged from the detection limit (<0.5
ng/samples) to approximately 50 ng/samples and cannot be determined directly by light
absorbance measurements. Accordingly, quantities of gDNA were estimated using the
PCR products after the first amplification step, and compared with the
microbial-particle concentrations that were determined by fluorescence microscopic
observation. The efficiency of the gDNA extraction from air samples was more than

80 %.

Before the analysis of bacterial community structures, USEARCH v.8.01623
(Edgar, 2013) was used to process the raw Illumina sequencing reads. Anomalous
sequences were removed with the following workflow. First, the forward and reverse
paired-end reads were merged, and the merged reads with lengths outside of the
200-500 bp range or those exceeding 6 homopolymers were discarded using Mothur
v1.36.1 (Schloss et al., 2009). Next, the sequences were subjected to Q-score filtering to
remove reads with more than one expected error. Reads occurring only once in the
entire dataset (singleton) were then removed. Theses sequences were clustered *de novo*
(with a minimum identity of 97 %) into 204 operational taxonomic units (OTUs) among
the 18 samples. The taxonomy of the representative OTU sequences was assigned using
the RDP classifier (Wang et al., 2007) implemented in QIME v1.9.1 (Caporaso et al.,
2010). Non-metric multidimensional scaling (NMDS) plot of the pairwise Bray-Curtis
distance matrix were used for the classification of all air samples. Greengenes release
13_8 (McDonald et al., 2012) was used as the reference taxonomic database.

*2.5. Accession numbers*
All data obtained from MiSeq sequencing data have been deposited in the
DDBJ/EMBL/GenBank database (accession number of the submission is
PRJEB17915).

**3. Results**

*3.1. Air mass analyses using LIDAR measurements, back trajectories, and metrological*

*data*

The vertical distributions of the depolarization ratio determined by LIDAR

measurements were assessed for the four sampling events (March 19, 2013; March 20,

2015; April 28, 2013; and March 28, 2014). The depolarization ratio increased at the

altitude of 3,000 m on March 19, 2013 (Fig. 2a), while it decreased at the middle

altitude of 1,000 m. The air mass on March 20, 2015 showed high values of

depolarization ratio at altitudes of 2,500 m and 500 m, consistent with the vertical

distribution of non-spherical (mineral dust) particles over the Noto Peninsula (Fig. 2d).

A 3-day back trajectory analysis indicated that the air mass at 3,000 m on both sampling

dates came from the Asian desert region to the Noto Peninsula (Hakui) immediately

across the Sea of Japan (Fig. 3). These results indicated the dust event occurrence on

March 19, 2013 was specific to the upper altitude of 3,000 m, while the dust event on

March 20, 2015 occurred between the altitudes of 2,500 m and 500 m. Moreover,

samples collected on April 28, 2013 and March 28, 2014 exhibited low depolarization

ratio (Fig. 2b-c), and the air masses on these two sampling dates came from areas of

North Asia, including eastern Siberia (Fig. 3).

The air-sampling periods from the March 2014 time series (from the 23rd to the

29th of March 2014) and the March 2015 time series (from the 16th to the 21st of

March 2015) showed different patterns of depolarization ratio and air mass trajectory

roots between the two series (Figs. 4 and 5). Depolarization ratio from March 2014

maintained lower values (Fig. 4a) and the trajectory lines changed the roots from eastern

Siberia to the Korean Peninsula before surrounding the Japanese islands (Fig. 4c). In
contrast, the sampling period during March 2015 had substantially higher depolarization
ratio, indicating a strong presence of mineral dust particles (Fig. 5a), and air masses at
3,000 m consistently originated from the Asian desert regions (Fig. 5c).
Temperatures from March 19, 2013; April 28, 2013; March 28, 2014; and March 20,
2015 increased from approximately 290 K to approximately 300 K at middle altitudes
(500 m and 1,200 m) (Fig. 2). The temperature profile clearly indicated the presence of
a thin boundary under the upper altitudes (2,500 m and 3,000 m), which suggested that
there is a difference in air qualities between the middle and upper altitudes (Table 1).
During the March 2014 time series, temperatures dynamically changed at altitudes of
approximately 1,200 m, while those from the March 2015 time series (the 16th, 17th,
and 21st of March 2015) were stable at 1,200 m (Figs. S1 and S2). These results
indicate that the boundary layers were located at 1,200 m during the March 2014 time
series, whereas the tropospheric air transported by westerly winds was suspended at the
sampling altitudes (500 m and 1,200 m) used during the March 2015 time series.

*3.2. Vertical distributions and sequential variations of aerosol particles*
Aerosol particle concentrations from the ground level to the troposphere were
measured using OPC to compare the vertical distributions of aerosols from the four
sampling events. The OPC-measured particles on March 19, 2013 and March 20, 2015
maintained similar concentrations below the troposphere (Fig. 2ad), while the
concentrations on April 28, 2013 and March 28, 2014 decreased one or two orders of
magnitude between the troposphere and ground level (Fig. 2bc). At high altitudes (2,000
m to 2,500 m), the course particles (greater 1.0 µm) observed on March 19, 2013 and
March 20, 2015 were one or two orders of magnitude higher ($10^5$ to $10^6$ particles m$^{-3}$)
than those on April 28, 2013 and March 28, 2014 (no more than $1.2 \times 10^4$ particles m$^{-3}$).
The fine particles (0.3 µm to 1.0 µm) showed similar concentrations between the four
sampling events, fluctuating between $1.2 \times 10^6$ to $3.5 \times 10^7$ particles m$^{-3}$. At lower
altitudes (130 m to 510), the aerosol particles had similar concentrations and size
distributions between the four sampling periods; the course particle concentration
ranged from $8.4 \times 10^5$ particles m$^{-3}$ to $1.2 \times 10^6$ particles m$^{-3}$, and the fine particles
ranged from $1.3 \times 10^7$ particles m$^{-3}$ to $1.2 \times 10^8$ particles m$^{-3}$.

OPC measurements indicated that air samples collected at 1,200 m during the March

2015 time series consistently contained course particles at one or two orders of
magnitude higher in concentration ($1.4 \times 10^6$ to $3.4 \times 10^6$ particles m$^{-3}$) than detected in
the March 2014 time series, which had concentrations of no more than $1.8 \times 10^5$
particles m$^{-3}$ (Fig. 4b). The concentration of relatively large particles (>5.0 µm) in
March 2015 maintained relatively higher concentrations (from $1.4 \times 10^4$ to $8.2 \times 10^5$
particles m$^{-3}$) than those observed in March 2014 (no more than $3.74 \times 10^3$ particles m$^{-3}$).
In contrast, the fine particles measured in March 2014 and March 2015 fluctuated
around similar concentrations ranging from $10^7$ to $10^8$ particles m$^{-3}$.

Based on the above observations, the sampled air masses that were influenced by

Asian dust events and included dust particles were categorized as "dust samples". The
sampled air masses that were not influenced by dust events or contained less dust
particles were categorized as "non-dust samples", in relation to the presence or absence
of dust events as the source of the aerosol samples (Table 1).

*3.3. Fluorescent microscopic observation of aerosol particles*

Using epifluorescence microscopy with DAPI staining, the aerosol particles in the 18 air samples emitted several types of fluorescence, categorized as white, blue, yellow, or black (Fig. S3). White fluorescence particles, (white particles) were indicative of mineral particles originating from the sand or soil. Microbial (prokaryotic) particles stained with DAPI emitted blue fluorescence, forming coccoid- or bacilli-like particles with a diameter <3 μm. Yellow fluorescence particles (yellow particles) stained with DAPI were organic matter and ranged from 1.0 μm to 10 μm in diameter. Most of the yellow particles disappeared in the aerosol-particle suspending solutions after protease treatment, suggesting that the yellow particles consisted mainly of proteins. Black particles were indicative of an anthropogenic black carbon originating from East Asian regions, produced by biomass burning, industrial activities, and vehicle exhaust.

The dust samples from upper altitudes (2,500 m and 3,000 m) contained 5 to 100 times higher concentrations of microbial, organic, and white particles than the concentrations detected in the non-dust samples (Fig. 2). In the upper altitude dust samples, the concentration of mineral particles ranged from $7.77 \times 10^5$ particles m$^{-3}$ to $1.08 \times 10^6$ particles m$^{-3}$ (Fig. 2ad), whereas the concentrations of the non-dust samples ranged from $3.14 \times 10^4$ particles m$^{-3}$ to $1.48 \times 10^5$ particles m$^{-3}$ (Fig. 2bc). The microbial particles in the high altitude dust samples exhibited concentrations of approximately $1.5 \times 10^6$ particles m$^{-3}$ that were two orders of magnitude higher than in the non-dust samples (approximately $6.0 \times 10^4$ particles m$^{-3}$). The organic particles in the high altitude dust samples were also found at higher concentrations of

approximately $4.2 \times 10^6$ particles m$^{-3}$ than those from the non-dust samples 13H428-u
and 14H328-u, which were $2.12 \times 10^4$ particles m$^{-3}$ and $5.30 \times 10^4$ particles m$^{-3}$,
respectively. In contrast, the air samples collected at the low altitude of 10 m exhibited a
random or stochastic pattern between $10^5$ and $10^6$ particles m$^{-3}$, regardless of the
sampling dates (Fig. 2). Black particles were observed in the four air samples from 10 m
and fluctuated around concentrations of less than $8.48 \times 10^4$ particles m$^{-3}$. Finally, the
percentage of organic particles out of the total number of particles (organic and
microbial particles) in the dust samples 13H319-u, 15H320-u, and 15H320-m ranged
between approximately 71.5 % and 73.6 %, which was higher than in the non-dust
samples, which ranged from 4.6 % to 46.3 % (Fig. S4).
All types of fluorescence particles were also observed in the sequentially collected
air samples at 1,200 m in the March 2015 time series (except for 2,500 m on March
20th) and the March 2014 series. The dust samples examined from the March 2015
series had higher concentrations of total particles than the non-dust samples of the
March 2014 series (Figs. 4 and 5). The mineral particles detected in the March 2014
series fluctuated at low concentrations from $3.39 \times 10^4$ particles m$^{-3}$ to $2.62 \times 10^5$
particles m$^{-3}$ (Fig. 4), while in the March 2015 series the mineral particles showed
higher values from $1.80 \times 10^5$ particles m$^{-3}$ to $1.77 \times 10^7$ particles m$^{-3}$ (Fig. 5). High
levels of organic particles were detected in the March 2015 series samples, ranging from
$3.13 \times 10^5$ to $3.75 \times 10^7$ particles m$^{-3}$, which decreased to below $2.28 \times 10^5$ particles m$^{-3}$
in the March 2014 series samples. The microbial particle concentrations in the March
2015 series samples (ranging from $4.75 \times 10^5$ to $2.06 \times 10^6$ particles m$^{-3}$) were higher
than those of in the March 2014 series samples (ranging from $3.31 \times 10^5$ to $1.25 \times 10^6$
particles m$^{-3}$). The ratio of organic particles to the total number of organic and microbial
particles detected during March 2015 (71.5 % to 95.6 %) were higher than those during
March 2014 series (8.0 % to 36.2 %) (Fig. S4). The black particles were randomly
observed in all samples from March 2015 and March 2014.

*3.4. Analysis of bacterial communities using MiSeq sequencing analysis*
For the analysis of the prokaryotic composition in the 18 samples, we obtained
645,075 merged paired-end sequences with the lengths ranging from 244 bp to 298 bp
after quality filtering, and the sequence library size for each sample was normalized at
1,500 reads. The 16S rDNA sequences were divided into 204 phylotypes (sequences
with >97 % similarity). Phylogenetic assignment of sequences resulted in an overall
diversity of 16 phyla and candidate divisions, 32 classes (and class-level candidate taxa),
and 72 families (and family-level candidate taxa). The majority (>90 %) of the
sequences were represented by 9 bacterial classes and 33 families (Figs. 6 and 7). The
bacterial compositions varied during the sampling periods and included the phylotypes
belonging to the classes Cyanobacteria, Actinobacteria, Bacilli, Bacteroidetes, SBRH58,
and Proteobacteria (Alpha, Beta, Gamma, and Deltaproteobacteria), which are typically
generated from atmospheric, terrestrial and marine environments. On the box plots, the
numbers of bacterial species estimated by Chao I were similar at average levels between
the dust samples and non-dust samples, while the Chao I and Shannon values of the
non-dust samples showed a wider range than that of dust samples (Fig. 8a). A
non-metric multidimensional scaling (NMDS) plot demonstrated the distinct clustering
of prokaryotic communities separating the dust samples and the non-dust samples (Fig.
8b). For the PCR-analysis steps, negative controls (no template and template from
unused filters) did not contain 16S rDNA amplicons demonstrating the absence of
artificial contamination during experimental processes.

*3.5. Vertical distributions of bacterial communities in dust and non-dust samples*
The vertical distributions of bacterial compositions showed different patterns
between dust event days and non-dust days (Fig. 6). In the dust samples collected at
upper altitudes, phylotypes belonging to the phylum Bacilli accounted for more than
60.5 % of the total and were mainly composed of members of the families Bacillaceae
and Paenibacilliaceae (Fig. 6). Bacterial numbers from the phylum Bacilli decreased at
lower altitudes during dust events, and the phylotypes of Cyanobacteria, Actinobacteria,
and Protobacteria increased in relative abundance in the samples collected at middle and
low altitudes (13H319-m, 13H319-l, and 15H320-m).
Cyanobacteria, Actinobacteria, and Proteobacteria sequences also dominated in
the air samples collected during non-dust events (13H428-m, 14H328-u, 14H328-m,
and 14H328-l). Specifically, Actinobacteria phylotypes increased in their relative
abundance, ranging from 14.1 % to 24.7 % in the non-dust samples collected on March
28, 2014. Proteobacteria phylotypes containing several bacterial families occupied a
high relative abundance, ranging from 60.5 % to 85.3 % in the non-dust samples
13H428-u, 13H428-m, 14H328-u, 14H328-m, and 14H328-l. In particular, the non-dust
samples collected on March 28, 2014 included the Alphaproteobacteria phylotypes,
which have composed of members of the families Phyllobacteriaceae and
Sphingomonadaceae. Most Betaproteobacteria, phylotypes including the families
Oxalobacteraceae and Comamonadaceae, were specific to the non-dust samples
collected at 1,200 m and 2,500 m on April 28, 2013.
Cyanobacteria phylotypes, which were randomly detected from both dust samples
and non-dust samples, particularly increased in both the non-dust sample collected at 10
m on April 28, 2013 and the dust sample collected at 3,000 m on March 20, 2015, with
a relative abundance of 15.3 % and 74.6 %, respectively. Bacteroidia phylotypes also
randomly appeared in several air samples, regardless of the dust event influences and
were present at maximal levels in the non-dust sample 13H319-m, with a relative
abundance of 35.6 %.

*3.6. Variations in bacterial communities during dust events and non-dust events*
Sequential variations in the bacterial composition of air samples at altitudes of
1,200 m or 2,500 m were compared between dust event periods (March 2015 series) and
non-dust periods (March 2014 series). During the March 2015 dust event, phylotypes of
the family Bacillaceae in the class Bacilli occupied more than 53.0 % of the relative
abundance in the four dust samples collected (Fig. 7). Cyanobacteria phylotypes related
to the marine cyanobacterium Synechococcaceae uniquely appeared in the dust samples
of the March 2015 series; their abundance fluctuated the values ranging from 12.5 % to
14.8 % between the 16th and the 20th of March 2015 before decreasing to 1.5 % on
March 20.
During the non-dust periods of the March 2014 series at the middle altitude, the
relative abundance of Actinobacteria phylotypes belonging to the family
Micrococcaceae was occupied 59.9 % on March 23, decreased to 19.5 % on March 24,
and disappeared from samples collected on March 29. Corresponding to the decrease in
Actinobacteria phylotypes, Alpha and Gammaproteobacteria phylotypes showed an
increasing trend from 30.6 % to 96.8 % between the 23rd and the 29th of March 2014
(Fig. 7a). Alphaproteobacteria phylotypes belonging to the families
Sphingomonadaceae, and Phyllobacteriaceae, consistently appeared throughout the
sampling periods of the March 2014 series and occupied a maximum relative abundance
of 72.9 % and 22.3 % respectively. For Gammaproteobacteria, the Xanthomonadaceae
sequences dominated at a relative abundance of 18.3 % and 5.4 % in the non-dust
samples 14H325-m and 14H329-m, respectively, during the air mass was suspended the
Japanese islands for a few days.

**4. Discussion**

*4.1 Air mass conditions during Asian dust and non-dust events*

Westerly winds blowing over East Asia disperse airborne microorganisms

associated with dust mineral particles (Maki et al., 2008) and anthropogenic particles
(Cao et al., 2014; Wei et al., 2016), influencing the abundances and taxon compositions
of airborne bacteria at high altitudes over downwind areas, such as Noto Peninsula
(Maki et al., 2013). In this investigation, the increases in aerosol particles (dust
particles) and associated microbial particles were observed over the Noto Peninsula
during the dust events of March 19, 2013 and March 20, 2015 (Figs. 2 and 4). At the
two sampling dates, the air mass including microbial particles had traveled from the
Asian desert region throughout the anthropogenic polluted areas (Fig. 2), and the dust
particles entered the Japanese troposphere and were maintained at high altitudes (March
19, 2013) or mixed with the ground-surface air (March 20, 2015). During non-dust days,
the air masses at high altitudes came from several areas, including the eastern region of
Siberia, Asian continental coasts (Korean Peninsula), the Sea of Japan, or surrounding
Japanese islands, and mixed with ground-surface air over the Noto Peninsula. The air
samples collected during dust and non-dust events were valuable for understanding the
westerly wind influences on vertical distributions and sequential dynamics of airborne
bacteria at high altitudes over the downwind regions.

*4.2 Aerosol dynamics during Asian dust and non-dust event*

The microscopic fluorescence particles of all samples could be separated into four

categories: mineral (white), microbial (blue), organic (yellow), and black-carbon (black)
particles (Fig. S3), which were observed in the previous air samples collected during
dust events (Maki et al., 2015). The amount of microbial particles increased at high
altitudes during dust events, suggesting that the dust events directly carried bacterial
particles to the troposphere over downwind areas. At low altitudes, similar
concentrations of fluorescent particles were observed in air samples collected between
dust events (13H319-l) and non-dust events (13H428-l) (Fig. 2) because the dust
particles did not reach the ground surface on the dust-event days. In the absence of the
influences of dust-events, the aerosols mainly originated from local environments in
Japanese areas.

Organic particles also increased during dust events and in the ratios between all

particles related to the dust events. The organic particles originate from proteins and
other biological components (Mostajir et al., 1995). The tropospheric aerosols would be
composed of organic particles at high rates ranging from 30 % to 80 % (Jaenicke, 2005),
and organic particle concentrations fluctuated from $10^3$ to $10^5$ particles m$^{-3}$ at high
altitudes of 4,000 m above the ground (Twohy et al., 2016). The dead-phase cells of
microbial isolates obtained from aerosol samples mainly irradiated yellow fluorescence
instead of blue fluorescence (Liu et al., 2014). When fungi (*Bjerkandera adusta*) and
bacteria (*Bacillus* spp.) isolated from aerosol samples were incubated, the dead-phase
microbial cells mainly irradiated yellow fluorescence instead of blue fluorescence (Liu
et al., 2014; Fig. S3). The relative numbers of organic particles to the total number of
microbial and organic particles in the dust samples showed significantly higher values
(82.9 ± 32.3 %) than in the non-dust samples (23.3 ± 13.7 %) (Fig. S4). Hara and Zhang
reported that dust events in Kyushu, Japan, resulted in an increased ratio of damaged
microbial cells in the air at the ground-surface and that the ratio increased to
approximately 80 % (Hara and Zhang, 2012). Furthermore, organic molecules
associated with dust aerosols are reported to be composed of mannitol, glucose, and
fructose, which are part of cell components of airborne microorganisms and contribute
to the formation of secondary organic aerosols (SOA) (Fu et al., 2016). Microbial cells
or their components coming from Asian continents to Japan would be exposed to air at
high-altitudes during their long-range transport, increasing the ratios of damaged and
dead cells or SOA.

The appearance of black carbon most likely originated from anthropogenic

activities, such as biomass burning, industrial activities, and vehicle exhaust (Chung and
Kim, 2008). In the anthropogenic regions of eastern China, anthropogenic particles
originating from human activities are expected to comprise more than 90 % of dust
particles (Huang et al., 2015a). When the westerly winds are strongly blowing over the
Noto Peninsula, the black carbon particles at upper altitudes (3,000 m) are thought to
mainly derive from continental anthropogenic regions.

*4.3 Comparing the community structures of atmospheric bacteria between Asian dust*
*and non-dust events*
Dust events and air-pollutant occurrences changed the airborne bacterial
communities over the downwind areas, such as Beijing (Jeon et al., 2011; Cao et al.,
2014) and east Mediterranean areas (Mazar et al., 2016). The westerly winds blowing
over East Asia would transport airborne bacteria to the high-altitude atmosphere over
the Noto Peninsula (Maki et al., 2015) and North American mountains (Smith et al.,
2013). Our box plots analysis suggested that changes in the bacterial diversity in the
dust samples would be more stable than in the non-dust samples (Fig. 8a). Furthermore,
using a NMDS plot, the bacterial compositions in the dust samples could be
distinguished from non-dust samples (Fig. 8b). Thus, the aerosol particles transported
by Asian dust events changed the atmospheric bacterial composition at higher altitudes
over downwind areas.
The phylotypes in the dust samples were predominately clustered into the class
Bacilli (Fig. 4a), while the non-dust samples mainly included the phylotypes of the
classes Alpha, Beta, and Gammaproteobacteria and Actinobacteria. Our previous
investigations indicated that the bacterial communities at an altitude of 3,000 m over the
Noto Peninsula included more than 300 phylotypes, which were predominantly
composed of Bacilli phylotypes (Maki et al., 2015). Bacterial groups belonging to
Bacilli, Proteobacteria, and Actinobacteria have been reported as airborne bacteria
around European mountains (Vaïtilingom et al., 2012) as well as over Asian rural
regions (Woo et al., 2013). Some Bacilli isolates were found to act as ice-nucleating
agents and to be involved in ice cloud (Matulova et al., 2014; Mortazavi et al., 2015).
Isolates of Gammaproteobacteria isolates were obtained from mineral dust particles
(Hara et al., 2016a), glaciated high-altitude clouds (Sheridan et al., 2003), and plant
bodies (Morris et al., 2008), and some isolate species, such as *Pseudomonas*, were
confirmed to have the ice-nucleation activity. Accordingly, Bacilli and Proteobacteria
members associated with dust events could potentially contribute to climate change
resulting from dust events.

*4.4 Dominant bacterial populations in the air masses transported from Asian continents*
In some dust-event samples collected at high altitudes (13H319-u, 15H320-u, and
15H320-m), Bacilli sequences accounted for more than 52.7 % of the total number of
sequences (Fig. 6). Back trajectories on March 19, 2013 and March 20, 2015 came from
the Asian desert region to the Noto Peninsula. Some *Bacillus* species were
predominantly detected at high altitudes above the Taklimakan Desert (Maki et al.,
2008) and above downwind areas during Asian dust events (Maki et al., 2010 and 2013;
Smith et al., 2013; Jeon et al., 2011; Tanaka et al., 2011). *Bacillus* species are the most
prevalent isolates obtained from mineral dust particles collected over downwind areas
(Hua et al., 2007; Gorbushina et al., 2007).
Bacilli members can form resistant endospores that support their survival in the
atmosphere (Nicholson et al., 2000). The *Bacillus* isolates obtained from atmospheric
samples showed higher-level resistance to UV irradiation than normal isolates
(Kobayashi et al., 2015). In the Gobi Desert, dust events increase the diversity of
airborne microbial communities; after dust events, spore-forming bacteria, such as
*Bacillus*, increase in their relative abundances (Maki et al., 2016). Accordingly, in the
atmosphere, selected Bacilli members associated with dust particles would be
transported over long distances.
The Bacilli sequences showed different vertical variations between the two dust
events on March 19, 2013 and March 20, 2015. On March 19, 2013 (13H319-m), the
relative abundances of Bacilli sequences decreased dynamically from 3,000 m to 1,200
m, while unstable atmospheric layers on March 20, 2015 most likely mixed the
long-range transported bacteria with the regional bacteria over the Noto Peninsula. A
previous investigation also demonstrated the vertical mixture of airborne bacteria over
Suzu in the Noto Peninsula (Maki et al., 2010).
Actinobacteria sequences decreased in relative abundance between the 23rd and
29th of March 2014 corresponding with changes in the air mass trajectory roots from
north Asian regions, such as eastern Siberia and Japan (Fig. 7). Furthermore,
Actinobacteria sequences appeared in the samples collected from air masses that were
transported throughout the Korean Peninsula on March 19, 2013; April 28, 2013; and
March 20, 2015. Actinobacteria members are frequently dominant in terrestrial
environments but seldom survive in the atmosphere for a long time, because they cannot
form spores (Puspitasari et al., 2015). However, the family Micrococcaceae in
Actinobacteria was primarily detected from anthropogenic particles collected in Beijing,
China (Cao et al., 2014). Over anthropogenic source regions for Asian continents,
anthropogenic particles occupy more than 90 % of dust particles and originate from
soils disturbed by human activities in cropland, pastureland, and urbanized regions
(Huang et al., 2015a; Guan et al., 2016). Air masses transported from the continental
coasts are expected to include a relatively high abundance of Actinobacteria members
associated with anthropogenic particles.
Natural dust particles from Asian desert areas (Taklimkan and Gobi Deserts) are
transported in the free troposphere (Iwasaka et al., 1988) and vertically mixed with
anthropogenic particles during the transportation processes (Huang et al., 2015a). In
some cases, short-range transport of air masses would carry only anthropogenic
particles to Japan, because the anthropogenic particles are often dominant in Asian
continental coasts (Huang et al., 2015a). Actinobacteria members may have been
transported with anthropogenic particles from continental coasts.

*4.5 Dominant bacterial populations in the air masses originated from marine*
*environments and Japanese islands*
Proteobacteria sequences increased in their relative abundances at high altitudes
during non-dust sampling dates (13H428-u, 13H428-m, 14H328-u, 14H328-m, and
March 2014 series), when air mass origins at 1,200 m changed from the Korean
Peninsula to Japan (Fig. 7). Proteobacteria members were the dominate species in the
atmosphere over mountains (Bowers et al., 2012; Vaïtilingom et al., 2012; Temkiv et al.,
2012), in the air samples collected on a tower (Fahlgren et al., 2010), and from the
troposphere (DeLeon-Rodriguez et al., 2013; Kourtev et al., 2011). In the phylum
proteobacteria, the families Phyllobacteriaceae, Methylobacteriaceae, and
Xanthomonadaceae were predominately detected from the non-dust samples and are
associated with plant bodies or surfaces (Mantelin et al., 2006; Fürnkranz et al., 2008;
Khan and Doty, 2009; Fierer and Lennon, 2011). The Betaproteobacteria sequences in
the non-dust samples mainly contained the Oxalobacteraceae and Comamonadaceae
families, which are commonly dominate in freshwater environments (Nold and Zwart,
1998) as well as on plant leaves (Redford et al., 2010). In addition, the class
Alphaproteobacteria in the non-dust samples also included marine bacterial sequences
belonging to the family Sphingomonadaceae (Cavicchioli et al., 2003). Bacterial
populations originating from marine areas are prevalent in cloud droplets (Amato et al.,
2007), in air samples collected from the seashores of Europe (Polymenakou et al., 2008),
in storming troposphere (DeLeon-Rodriguez et al., 2013), and at high altitudes in
Japanese regions (Maki et al., 2014), suggesting that the marine environments represent
a major source of bacteria in clouds. The air masses suspended over the Sea of Japan or
Japanese islands during non-dust events (the March 2014 series) could include a high
relative abundance of airborne bacteria, which were transported from the surface-level
air over the marine environments and the regional phyllosphere.

*4.6. Bacterial populations commonly detected during dust events and the non-dust*
*events*

Sequences originating from Synechococcaceae (in the class Cyanobacteria)

randomly appeared in the MiSeq sequencing databases results obtained from air samples,
regardless of dust event occurrences. *Synechococcus* species in the family
Synechococcaceae can eliminate excess peroxide from photosynthesis to resist UV
radiation and oxygenic stress (Latifi et al., 2009), suggesting that these bacteria resist
environmental stressors in the atmosphere. In a previous study, the air samples
transported from marine environments to Japan predominately contained *Synechococcus*
species (Maki et al., 2014), which were dominant marine bacteria in the Sea of Japan
and the East China Sea (Choi and Noh, 2009). The cloud water at approximately 3,000
m above ground level was also dominated by Cyanobacteria populations, indicating
their atmospheric transport (Kourtev et al., 2011). In addition to Alphaproteobacteria,
marine cyanobacterial cells can be transported from seawater to the atmosphere, thereby
contributing to the airborne bacterial variations over the Noto Peninsula. Marine
bioaerosols originated from cyanobacteria and gram-negative bacteria (including
Alphaproteobacteria) are reported to contribute the increase of endotoxin levels in
coastal areas influencing human health by inflammation and allergic reaction
(Lang-Yona et al., 2014).
Bacteroidetes sequences were detected in some air samples collected during Asian
dust and non-dust events. Members of the phylum Bacteroidetes, which were composed
of the families Cytophagaceae, associate with organic particles in terrestrial and aquatic
environments (Turnbaugh et al., 2011; Newton et al., 2011). Furthermore, these
bacterial populations dominate the atmosphere and sand of desert areas, where plant
bodies and animal feces are sparsely present (Maki et al., 2016). These bacterial groups
possibly originated from organic-rich microenvironments in several areas, such as desert
and marine areas.

**5. Conclusion**

Air samples including airborne bacteria were sequentially collected at high altitudes over the Noto Peninsula during dust events and non-dust events. The sampled air masses could be categorized based on sample types with (dust samples) and without (non-dust samples) dust event influences. Bacterial communities in the air samples displayed different compositions between dust events and non-dust events. The dust samples were dominated by terrestrial bacteria, such as Bacilli, which are thought to originate from the central desert regions of Asia, and the bacterial compositions were similar between the dust samples. In contrast, the air masses of non-dust samples came from several areas, including northern Asia, continental coasts, marine areas, and Japan regional areas, showing different variations in bacterial compositions between the sampling dates. Some scientists have attempted to apply airborne bacterial composition as tracers of air mass sources at ground level (Bowers et al., 2011; Mazar et al., 2016). In this study, the terrestrial bacteria, such as Bacilli and Actinobacteria members (Bottos et al., 2014), were dominant populations in the air samples transported from Asian continental areas. The air samples when the air mass was suspended around Japanese islands, mainly included the members of the classes Alpha (Phyllobacteriaceae and Methylobacteriaceae), Gamma, and Betaproteobacteria, which are commonly dominated in phyllosphere (Redford et al., 2010; Fierer and Lennon, 2011) or freshwater environments (Nold and Zwart, 1998). The atmospheric aerosols transported via marine areas include a high relative abundances of marine bacteria belonging to classes Cyanobacteria (Choi and Noh, 2009) and Alphaproteobacteria

(Sphingomonadaceae) (Cavicchioli et al., 2003). This study suggested that bacterial
compositions in the atmosphere can be used as air mass tracers, which could identify the
levels of mixed air masses transported from different sources.
However, one limitation of our investigation is that the number of samples
analyzed was not sufficient to cover entire changes in airborne bacteria at high altitudes
over the Noto Peninsula. Although the airborne bacterial composition during non-dust
periods was found to change dynamically, only a few types of variation were followed
in this investigation. In the future, greater numbers of samples, which are sequentially
collected at high altitudes using this sampling method, will need to be originated to
more accurately evaluate bioaerosol tracers. Since helicopter sampling procedures
require sophisticated techniques and are expensive, the sample numbers at high altitudes
are difficult to increase. Air sampling at high altitudes should be combined with
sequential ground-air sampling to advance the understanding of the influence of
westerly winds on airborne bacterial dynamics in downwind areas. Metagenomic
analyses and microbial culture experiments would also provide valuable information
about airborne microbial functions relating to ice-nucleation activities, chemical
metabolism, and pathogenic abilities.

**Acknowledgments**
We are thankful for the advice given by Dr. Richard C. Flagan of California Institute
of Technology and the sampling support from Dr. Atsushi Matsukia and Dr. Makiko
Kakikawa of Kanazawa University. Trajectories were produced by the NOAA Air
Resources Laboratory (ARL), which provided the HYSPLIT transport and dispersion
model and/or READY website (http://www.ready.noaa.gov). Members of Fasmac Co.,
Ltd. helped with the MiSeq sequencing analyses. This research was funded by the
Grant-in-Aid for Scientific Research (B) (No. 26304003) and (C) (No. 26340049). The
Bilateral Joint Research Projects from the Japan Society for the Promotion of Science
also supported this work, as did the Strategic International Collaborative Research
Program (SICORP: 7201006051) and Strategic Young Researcher Overseas Visits
Program for Accelerating Brain Circulation (No. G2702). This study was supported by
the Joint Research Program of Arid Land Research Center, Tottori University (No.
28C2015).

**Competing Interests**
The authors declare that they have no conflict of interest.

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

**Figure Captions**

Fig. 1. Sampling location (a) and helicopter flight routes during the sampling periods on
March 19, 2013, and April 28, 2013 (b); the 23rd, 24th, 25th, and 29th of March 2014
(c); and the 16th, 17th, 20th, and 21st of March 2015 (d).

Fig. 2. LIDAR observation of the depolarization ratio in Toyama city as well as vertical
changes in temperature, relative humidity, and potential temperature, and vertical
distributions of concentrations of OPC-counted particles and DAPI-stained particles
from the four sampling events on March 19, 2013 (a); April 28, 2013 (b); March 28,
2014 (c); and March 20, 2015 (d). The red circles in the LIDAR images indicate that the
sampling air included dust mineral particles (solid line) or that dust-event influences are
absent at the altitudes on the sampling time (dotted line). OPC-counted particles were
categorized according to diameter sizes of 0.3–0.5 μm (closed squares), 0.5–0.7 μm
(closed triangles), 0.7–1.0 μm (closed circles), 1.0–2.0 μm (closed diamonds), 2.0–5.0
μm (crosses), and >5.0 μm (open circles). DAPI-stained particles were classified into
microbial particles (blue bars), white particles (white bars), yellow fluorescent particles
(yellow bars), and black carbon (gray bars).

Fig. 3. Trajectories 3 days ago of aerosols that arrived at 2,500 m (blue-type lines) and
1,200 m (red-type lines) in Hakui, Japan, every hour for 5 h before the completion of
sampling time at the four dates; March 19, 2013; April 28, 2013; March 28, 2014; and
March 20, 2015.

Fig. 4. (a) LIDAR observation of the depolarization ratio in Toyama city and
concentrations of OPC-counted particles and DAPI-stained particles during no-dust
days from 0:00 UTC on March 23 to 0:00 UTC on March 30, 2014. The red circles with
dotted lines in the LIDAR images indicate dust-event influences are absent at the
altitudes on the sampling time. (b) OPC-counted particles were categorized according to
diameter sizes of 0.3–0.5 μm (closed squares), 0.5–0.7 μm (closed triangles), 0.7–1.0
μm (closed circles), 1.0–2.0 μm (closed diamonds), 2.0–5.0 μm (crosses), and >5.0 μm
(open circles). DAPI-stained particles were classified into microbial particles (blue bars),
white particles (white bars), yellow particles (yellow bars), and black particles (gray
bars). (c) Trajectories 3 days ago of aerosols that arrived at 2,500 m (blue-type lines)
and 1,200 m (red-type lines) in Hakui, Japan, every hour for 5 h before the completion
of sampling time during sampling periods on the 23rd, 24th, 25th, 28th, and 29th of
March 2014.

Fig. 5. (a) LIDAR observation of the depolarization ratio in Toyama city and
concentrations of OPC-counted particles and DAPI-stained particles during dust event
days from 0:00 UTC on March 16 to 0:00 UTC on March 23, 2015. The red circles with
solid lines in the LIDAR images indicate that the sampling air included dust mineral
particles. (b) OPC-counted particles were categorized based on diameter sizes of
0.3–0.5 μm (closed squares), 0.5–0.7 μm (closed triangles), 0.7–1.0 μm (closed circles),
1.0–2.0 μm (closed diamonds), 2.0–5.0 μm (crosses), and >5.0 μm (open circles).
DAPI-stained particles were classified into microbial particles (blue bars), white
particles (white bars), yellow particles (yellow bars), and black particles (gray bars). (c)
Trajectories 3 days ago of aerosols that arrived at 2,500 m (blue-type lines) and 1,200 m
(red-type lines) in Hakui, Japan, every hour for 5 h before the completion of sampling
time during sampling periods on the 16th, 17th, 20th, and 21st of March 2015.

Fig. 6. Vertical variations in bacterial compositions at (a) the class level and (b) the
family level of the partial sequences obtained in the MiSeq sequencing database (ca.
400 bp) obtained from air samples collected at different altitudes over the Noto
Peninsula at dust-event days (March 19, 2013; March 20, 2015) and non-dust-event
days (March 19, 2013; March 20, 2015).

Fig. 7. Changes in bacterial compositions at (a) the class level and (b) the family level
of the partial sequences obtained in the MiSeq sequencing database (ca. 400 bp) from
air samples collected at altitudes of 1,200 m (except for the sample collected at 500 m
on March 20, 2015) over the Noto Peninsula during dust-event days from the 16th to the
23rd of March 2015 and during non-dust-event days from the 23rd to the 29th of March

1088  2014.


Fig. 8. Comparison of the bacterial compositions among all air samples collected over
the Noto Peninsula. (a) Box plots of Chao 1 and Shannon analyses indicating the
bacterial diversity observed in dust samples and non-dust samples. Species were binned
at the 97 % sequence similarity level. (b) NMDS of the pairwise Bray-Curtis distance
matrix displaying clustering by all the air samples. Red indicates the samples that were
collected during dust-events and blue indicates those collected during non-dust-events
as determined by meteorological data. Circle indicates that the sample contained dust
particles as identified via microscopic observation, and triangle indicates that dust
particles were absent from the sample. The confidence ellipses are based on a
multivariate t-distribution, and represents the 95 % confidence interval of the
occurrence of dust vs. non-dust events when the samples were collected.

**Table Captions**

Table 1 Sampling information during the sampling periods.

Table 2. Researches targeting bacterial communities associated with Asian-dust events.

**Table 1 Sampling information during the sampling periods.**

| Sample name | Sampling date | Collection time (JST) | Total time (min) | Air volume | Sampling method | Sampling location[*1] | Free troposhere[*2] | Dust event day[*3] | Dust influence[*4] |
|---|---|---|---|---|---|---|---|---|---|
| 13H319-u | 19 March 2013 | 14:04 − 15:04 | 60 | 700 L | helicopter | 2500m | FT | + | dust sample |
| 13H319-m | | 15:19 − 16:19 | 60 | 700 L | helicopter | 1200m | ABL | + | non-dust sample |
| 13H319-l | | 14:25 − 15:25 | 60 | 700 L | building | 10m | GL | + | non-dust sample |
| 13H428-u | 28 April 2013 | 12:10 − 13:04 | 56 | 653 L | helicopter | 2500m | FT | - | non-dust sample |
| 13H428-m | | 13:13 − 14:03 | 50 | 583 L | helicopter | 1200m | ABL | - | non-dust sample |
| 13H428-l | | 12:03 − 13:03 | 60 | 700 L | building | 10m | GL | - | non-dust sample |
| 14H328-u | 28 March 2014 | 12:50 − 13:50 | 60 | 700 L | helicopter | 3000m | FT | - | non-dust sample |
| 14H328-m | | 14:04 − 15:04 | 60 | 700 L | helicopter | 1200m | ABL | - | non-dust sample |
| 14H328-l | | 13:00 − 14:00 | 60 | 700 L | building | 10m | GL | - | non-dust sample |
| 15H320-u | 20 March 2015 | 12:26 − 13:23 | 47 | 548 L | helicopter | 2500m | FT | + | dust sample |
| 15H320-m | | 13:39 − 14:40 | 60 | 711 L | helicopter | 500m | ABL | + | dust sample |
| 14H323-m | 23 March 2014 | 10:45 − 11:02 | 17 | 11.1 L | helicopter | 1200m | ABL | - | non-dust sample |
| 14H324-m | 24 March 2014 | 9:09 − 9:30 | 21 | 13.7 L | helicopter | 1200m | ABL | - | non-dust sample |
| 14H325-m | 25 March 2014 | 9:31 − 9:50 | 29 | 18.9 L | helicopter | 1200m | ABL | - | non-dust sample |
| 14H328-m | 28 March 2014 | 14:04 − 15:04 | 60 | 700 L | helicopter | 1200m | ABL | - | non-dust sample |
| 14H329-m | 29 March 2014 | 9:06 − 9:24 | 15 | 9.75 L | helicopter | 1200m | PT | - | non-dust sample |
| 15H316-m | 16 March 2015 | 11:21 − 11:43 | 22 | 14.3 L | helicopter | 1200m | FT | + | dust sample |
| 15H317-m | 17 March 2015 | 11:04 − 11:31 | 27 | 17.6 L | helicopter | 1200m | FT | + | dust sample |
| 15H320-u | 20 March 2015 | 12:26 − 13:23 | 47 | 548 L | helicopter | 2500m | FT | + | dust sample |
| 15H321-m | 21 March 2015 | 15:35 − 15:55 | 20 | 13.0 L | helicopter | 1200m | FT | + | dust sample |

*1   Height above the ground.

*2   Free troposhere: FT, Atmospheric boundary layer: ABL, Phase transiens: PT, GL: Ground level

*3   The occurences of dust evnets are evaluated by depending on LIDAR data   or trjectories. Dust-event day: +, non-dust-event day: -

*4   The air sample including   dust particle (dust sample) or that without dust particles (non-dust sample) are identified via microscopic observation.

Table 2. Researches targeting bacterial communities associated with Asian-dust events

| Sampling area[*1] | Sample | Location | Altitudes (m) | Sampling place | Sampling method | Analytical method for microorganisms | Dominated bacteria[*2] | | | references |
|---|---|---|---|---|---|---|---|---|---|---|
| | | | | | | | 1st | 2nd | 3rd | |
| Dust source area | Soil | Taklamakan Desert, China | 0 | Ground surface | Soil sampling | Clone library | Bacteroidetes (Sphingobacteriia) | Actinobacteria ( Actinobacteria) | Proteobacteria (Alpha, Beta, Gamma) | Yamaguchi et al. 2012 |
| Dust source area | Soil | Gobi Desert, China | 0 | Ground surface | Soil sampling | Clone libarary | Actinobacteria ( Actinobacteria) | Proteobacteria (Beta) | Bacteroidetes (Sphingobacteriia) | Yamaguchi et al. 2012 |
| Dust source area | Soil | Taklamakan Desert, China | 0 | Ground surface | Soil sampling | Pyrosequencing | Firmicutes (Bacilli)† | Actinobacteria | Proteobacteria (Gamma) | An et al. 2013 |
| Dust source area | Soil | Gobi Desert, China | 0 | Ground surface | Soil sampling | Pyrosequencing | Firmicutes (Bacilli)† | Proteobacteria (Gamma) | Bacteroidetes | An et al. 2013 |
| Dust source area | Soil | Taklamakan Desert, China | 0 | Ground surface | soil samples | Clone library | Actinobacteria ( Actinobacteria) | Firmicutes (Bacilli) | Proteobacteria | Puspitasari et al. 2016 |
| Dust source and deposition areas | Soil | Loess plateau, China | 0 | Ground surface | Soil sampling | Clone library | Proteobacteria (Beta, Gamma) | Actinobacteria ( Actinobacteria) | Bacteroidetes (Sphingobacteriia) | Yamaguchi et al. 2012 |
| Dust source and deposition areas | Soil | Loess plateau, China | 0 | Ground surface | Soil sampling | PCR-DGEE | Proteobacteria | Bacteroidetes | Gemmatimonadetes | Kenzaki et al. 2010 |
| Dust source area | Air | Tsogt-Ovoo, Mongolia | 3 | Ground surface | Filtration | MiSeq sequencing | Proteobacteria (Alpha) | Firmicutes (Bacilli) | Actinobacteria ( Actinobacteria) | Maki et al. 2017 |
| Dust source area | Air | Dunhuang, China | 10 | Top of building | Filtration | Clone library | Firmicutes (Bacilli)† | Proteobacteria | Bacteroidetes | Puspitasari et al. 2016 |
| Dust source area | Air | Dunhuang, China | 800 | Balloon | Filtration | PCR-DGEE | Firmicutes (Bacilli)† | - | - | Maki et al. 2008 |
| Dust source area | Air | Dunhuang, China | 800 | Balloon | Filtration | Clone library | Proteobacteria (Gamma) | Firmicutes (Bacilli) | - | Kakikawa et al. 2009 |
| Dust deposition area | Air | Noto peninsula, Japan | 3000 | Aircraft | Filtration | Clone library | Firmicutes (Bacilli)† | Bacteroidetes (Bacteroidia) | Proteobacteria (Gamma) | Maki et al. 2013 |
| Dust deposition area | Air | Noto peninsula, Japan | 3000 | Aircraft | Filtration | MiSeq sequencing | Firmicutes (Bacilli)† | Actinobacteria ( Actinobacteria) | Proteobacteria (Alpha&Beta) | Maki et al. 2015 |
| Dust deposition area | Air | Mt. Bachelor, USA | 2700 | Mt. Bachelor | Filtration | Culture | Firmicutes (Bacilli)† | Actinobacteria ( Actinobacteria) | Proteobacteria (Gamma) | Smith et al. 2012 |
| Dust deposition area | Air | Mt. Bachelor, USA | 2700 | Mt. Bachelor | Filtration | Microarray | Proteobacteria (Beta&Gamma) | Actinobacteria ( Actinobacteria) | Firmicutes (Bacilli)† | Smith et al. 2013 |
| Dust deposition area | Snow | Mt. Tateyama, Japan | 2450 | Mt. Tateyama | Snow sampling | PCR-DGEE | Firmicutes (Bacilli)† | Proteobacteria (Beta, Gamma) | Actinobacteria ( Actinobacteria) | Tanaka et al. 2011 |
| Dust deposition area | Snow | Mt. Tateyama, Japan | 2450 | Mt. Tateyama | Snow sampling | PCR-DGEE | Firmicutes (Bacilli)† | Proteobacteria (Beta) | Actinobacteria ( Actinobacteria) | Maki et al. 2011 |
| Dust deposition area | Air | Noto peninsula, Japan | 1200 | Helicopter | Filtration | MiSeq sequencing | Firmicutes (Bacilli)† | Proteobacteria (Alpha, Gamma) | Cyanobacteria | This study |
| Dust deposition area | Air | Suzu, Japan | 1000 | Balloon | Filtration | MiSeq sequencing | Firmicutes (Bacilli)† | Proteobacteria (Alpha) | Deinococcus-Thermus (Deinococci) | Maki et al. 2015 |
| Dust deposition area | Air | Osaka, Japan | 900 | Air craft | Filtration | Clone library | Firmicutes (Bacilli) | Bacteroidetes (Sphingobacteriia) | Actinobacteria ( Actinobacteria) | Yamaguchi et al. 2012 |
| Dust deposition area | Air | Suzu, Japan | 800 | Balloon | Filtration | Clone library | Firmicutes (Bacilli)† | Bacteroidetes (Bacteroidia) | Proteobacteria (Gamma) | Maki et al. 2013 |
| Dust deposition area | Air | Suzu, Japan | 600 | Balloon | Filtration | PCR-DGEE | Firmicutes (Bacilli)† | - | - | Maki et al. 2010 |
| Dust deposition area | Air | Seoul, South Korea | 25 | Top of building | Liquid impiger | Pyrosequencing | Actinobacteria ( Actinobacteria) | Proteobacteria (Alpha, Gamma) | Firmicutes (Bacilli)† | Cha et al. 2017 |
| Dust deposition area | Air | Osaka, Japan | 20 | Top of building | Filtration | Pyrosequencing | Actinobacteria ( Actinobacteria) | Cyanobacteria | Acidobacteria (Acidobacteria) | Park et al. 2016 |
| Dust deposition area | Air | Seoul, South Korea | 17 | Top of building | Filtration | PCR-DGEE | Actinobacteria ( Actinobacteria) | Firmicutes (Bacilli)† | Proteobacteria (Gamma) | Lee et al. 2011 |
| Dust deposition area | Air | Beijing, China | 15 | Top of building | Filtration | Pyrosequencing | Firmicutes (Bacilli) | Proteobacteria (Gamma) | Bacteroidetes (Flavobacteriia) | Wei et al. 2016 |
| Dust deposition area | Air | Beijing, China | 10 | Top of building | Filtration | HiSeq sequencing | Actinobacteria ( Actinobacteria) | Proteobacteria (Alpha, Beta, Gamma) | Chloroflexi (Thermomicrobia) | Cao et al. 2014 |
| Dust deposition area | Air | Seoul, South Korea | 10 | Top of building | Filtration | Clone library | Firmicutes (Bacilli)† | Actinobacteria | Bacteroidetes | Jeon et al. 2011 |
| Dust deposition area | Air | Suzu, Japan | 10 | Top of building | Filtration | MiSeq sequencing | Firmicutes (Bacilli)† | Deinococcus-Thermus (Deinococci) | Proteobacteria (Alpha) | Maki et al. 2015 |
| Dust deposition area | Air | Goyang, South Korea | - | Top of building | Filtration | Pyrosequencing | Actinobacteria ( Actinobacteria) | Proteobacteria (Gamma) | Firmicutes (Bacilli)† | Cha et al. 2016 |
| Dust deposition area | Air | Kanazawa, Japan | 10 | Roof of building | Filtration | MiSeq sequencing | Firmicutes (Bacilli)† | Cyanobacteria | Proteobacteria (Alpha) | Maki et al. 2014 |
| Dust deposition area | Air | Pacific Ocean | - | Ship board | Filtration | Pyrosequencing | Firmicutes (Bacilli)† | Proteobacteria (Beta, Gamma) | Cyanobacteria | Xia et al. 2015 |

*1 Dust source area: the areas providing dust mineral particles, Dust deposition area: the area where the dust mineral paticles deposit

*2 The bacterial phyla in the orders of large abundance rates in each samples. † indaictes that Firmicutes (Bacilli) predominately included *Bacillus* members.

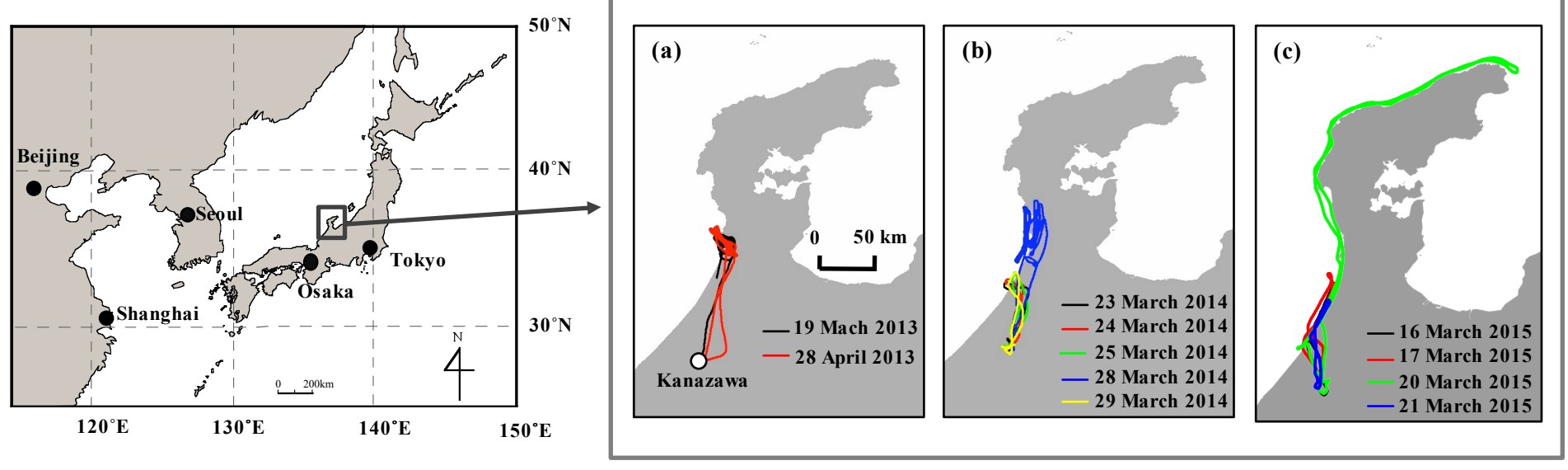

Fig. 1 T. Maki et al.

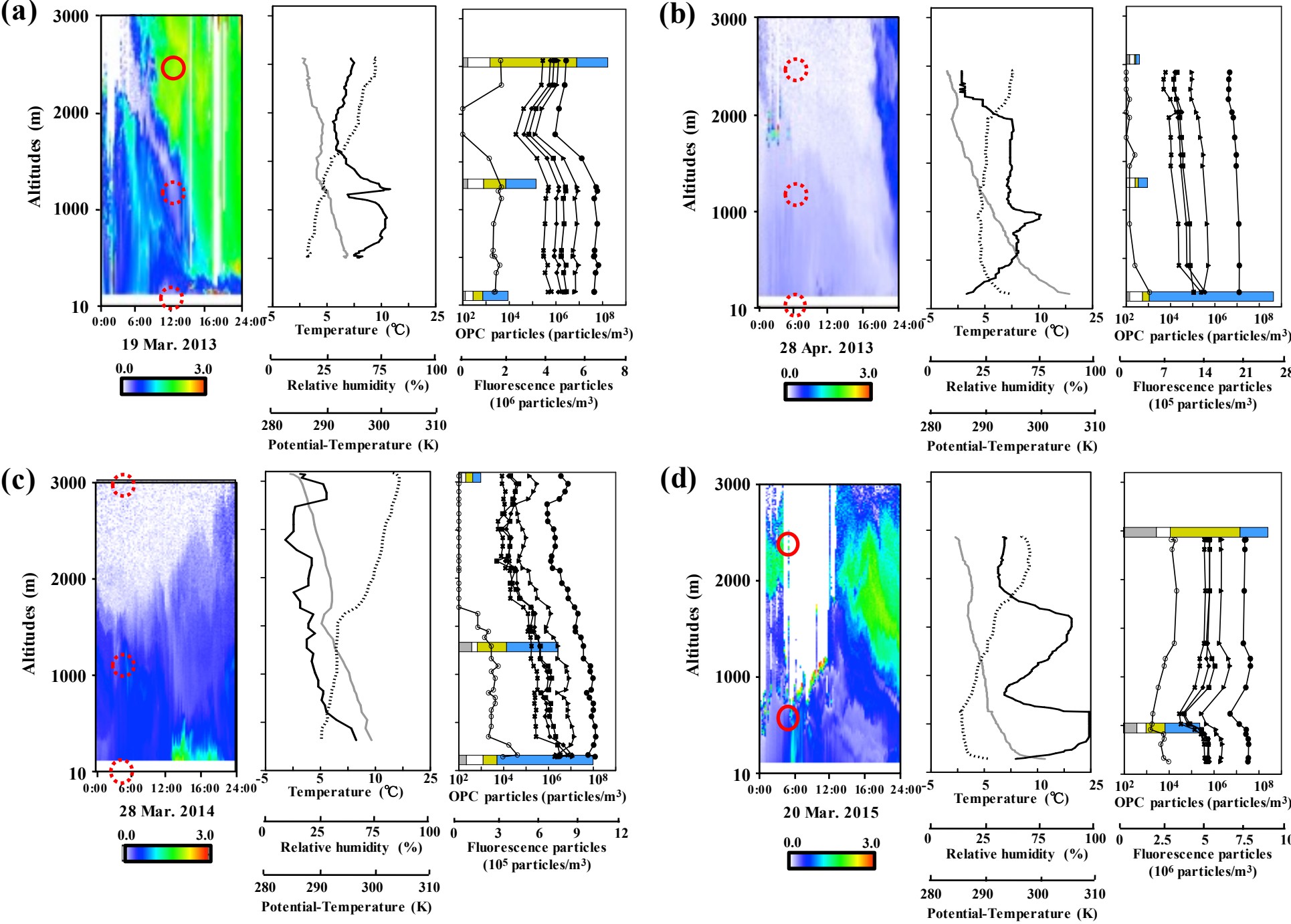

Fig. 2 T. Maki et al.

**19 Mar.
2013**

**28 Apr.
2013**

**28 Mar.
2014**

**20 Mar.
2015**

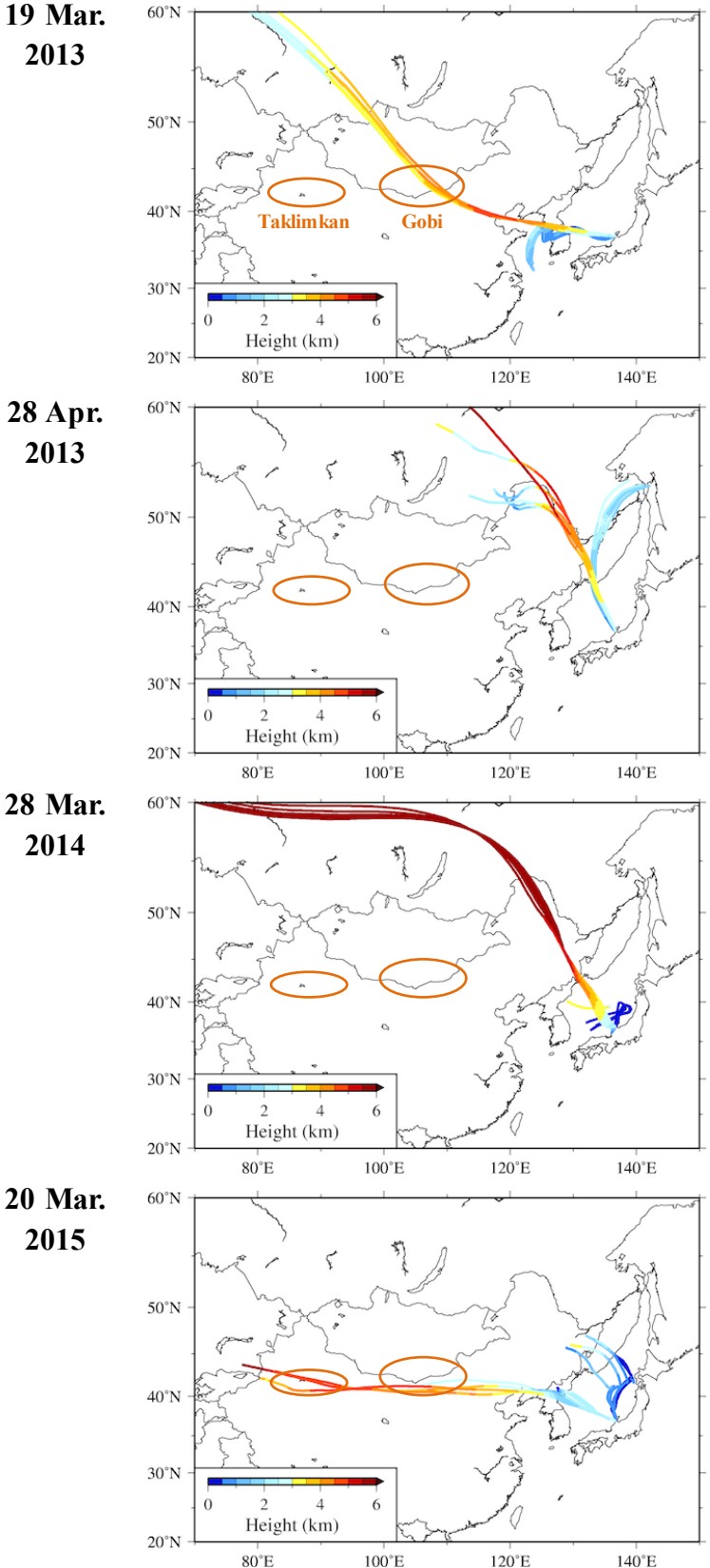

Fig. 3 T.Maki et al.

**(a)**

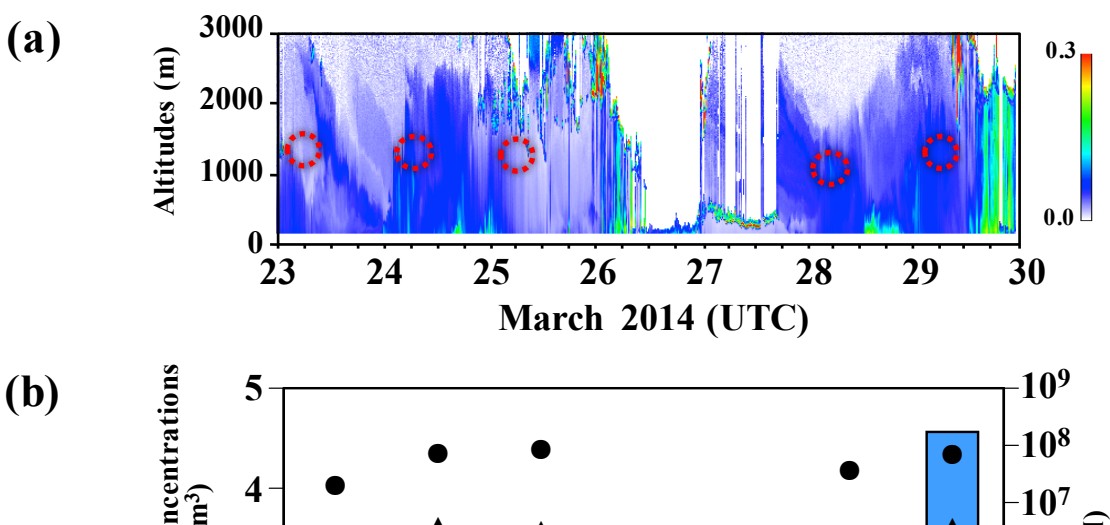

**(b)**

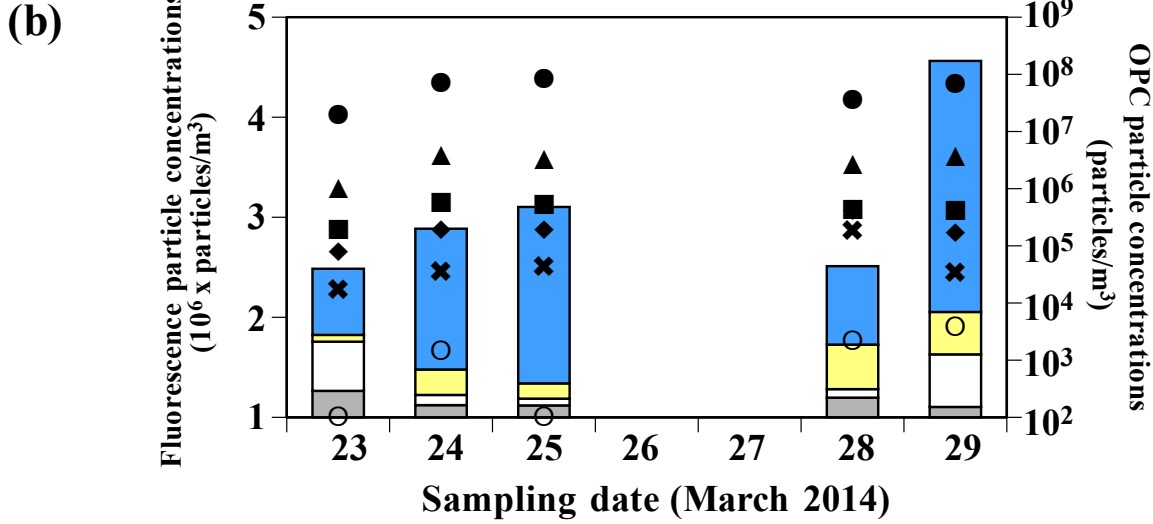

**(c)**

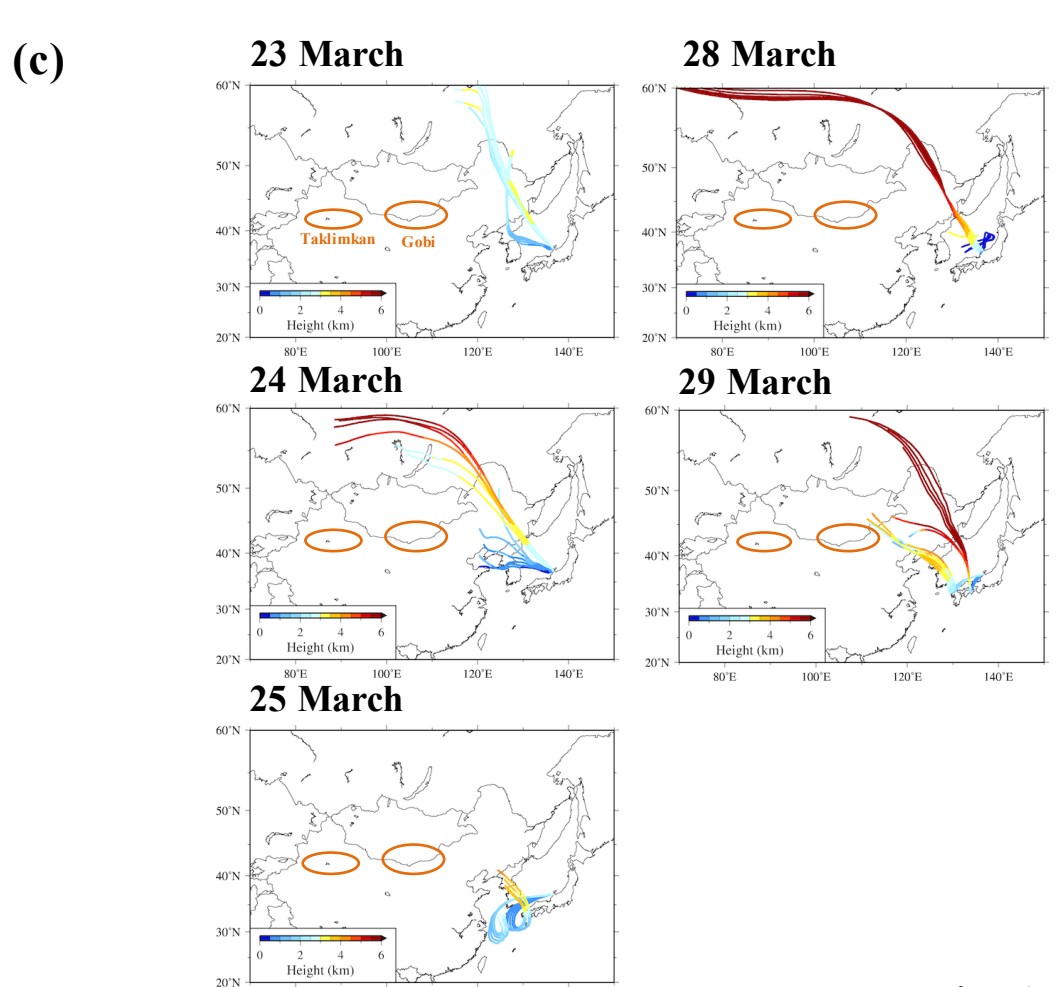

Fig. 4 T.Maki et al.

**(a)**

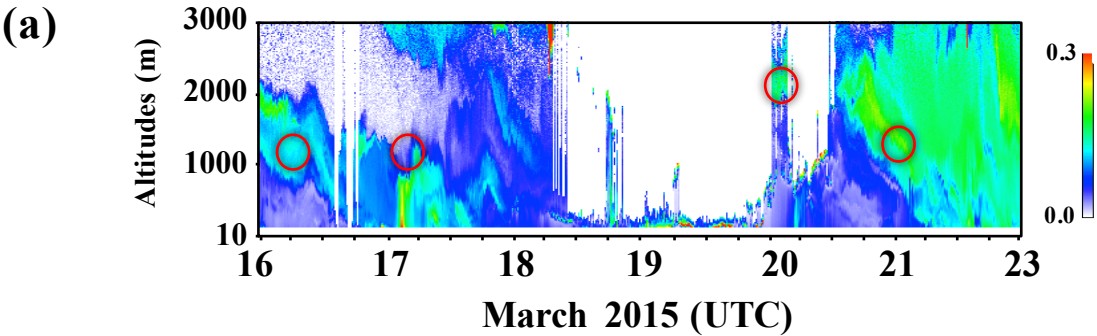

**(b)**

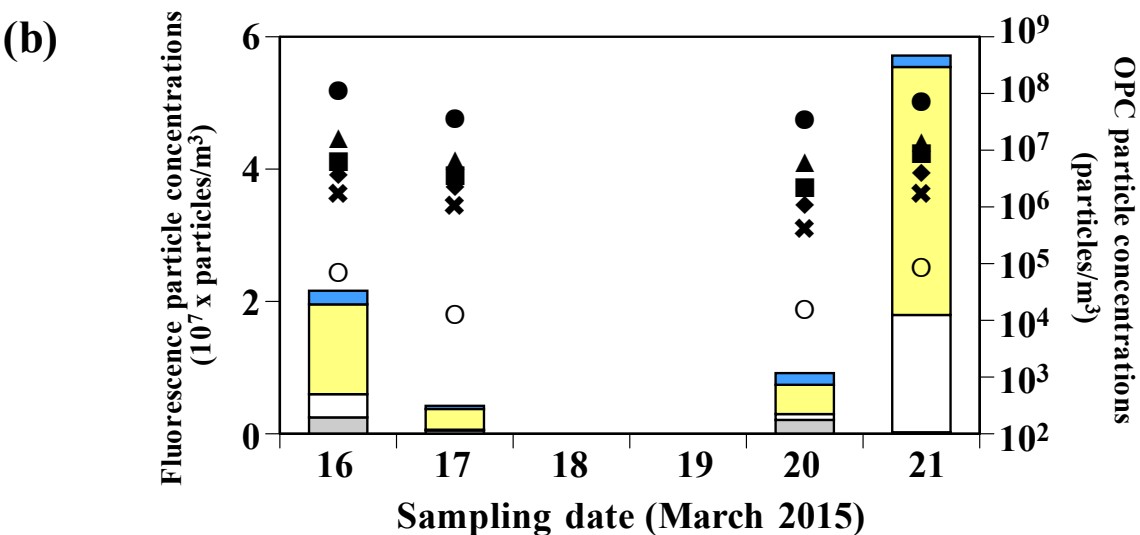

**(c)**

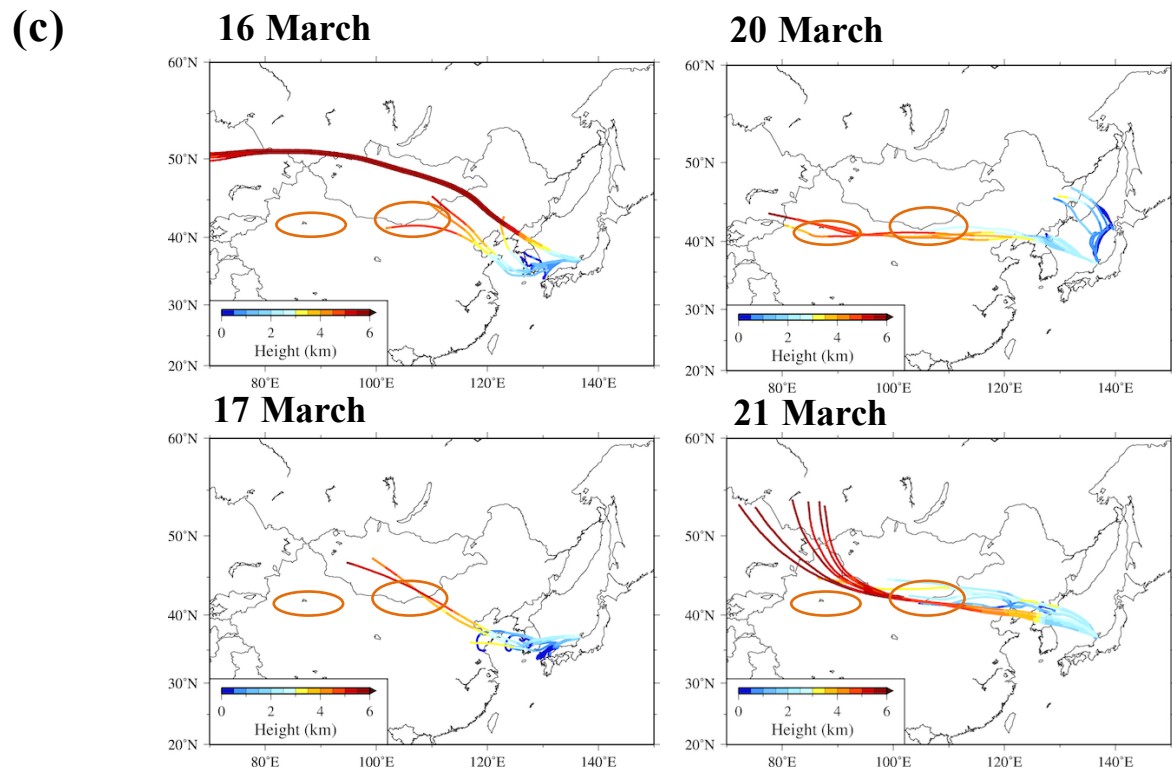

Fig. 5 T.Maki et al.

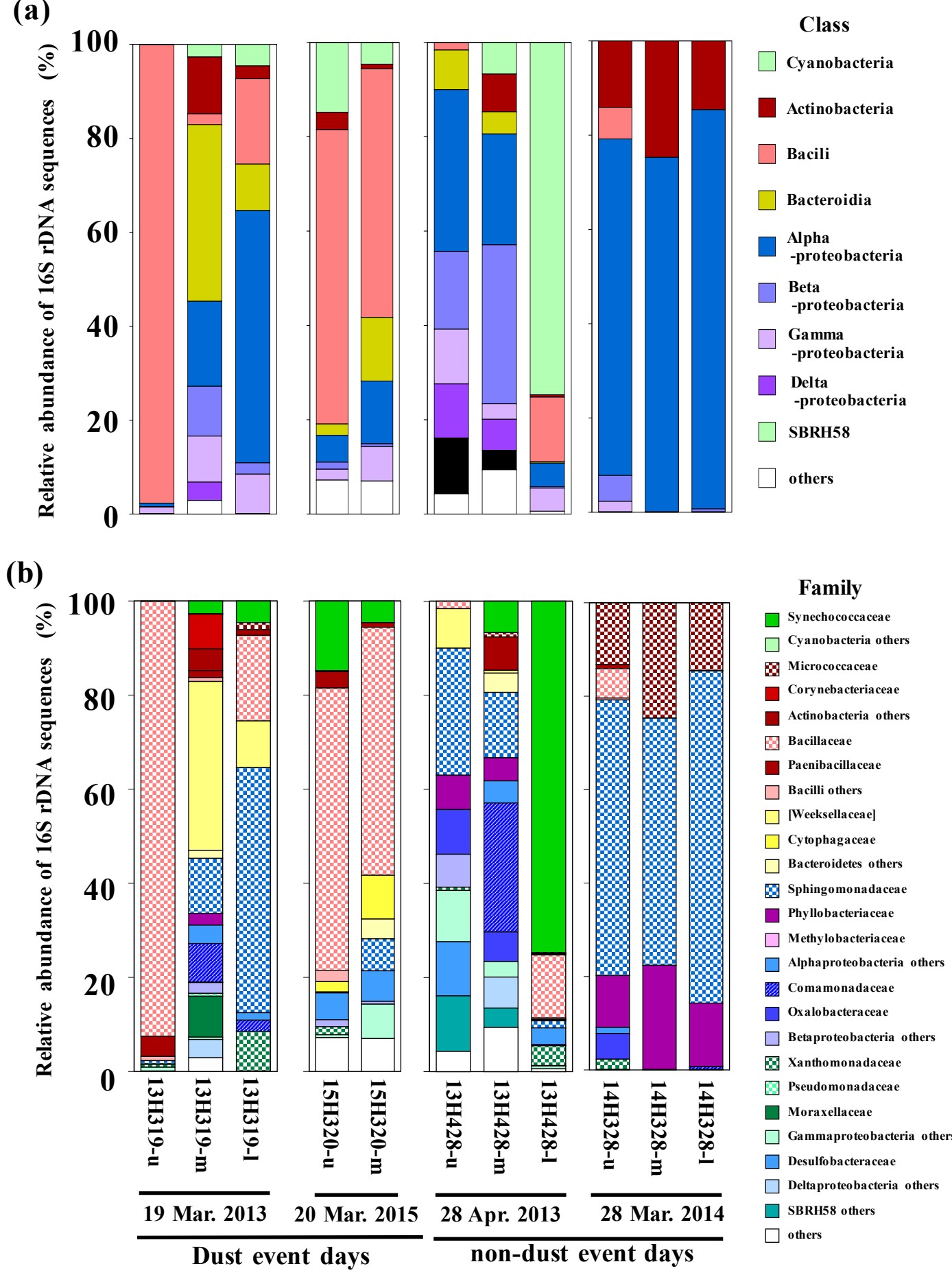

Fig. 6 T.Maki et al.

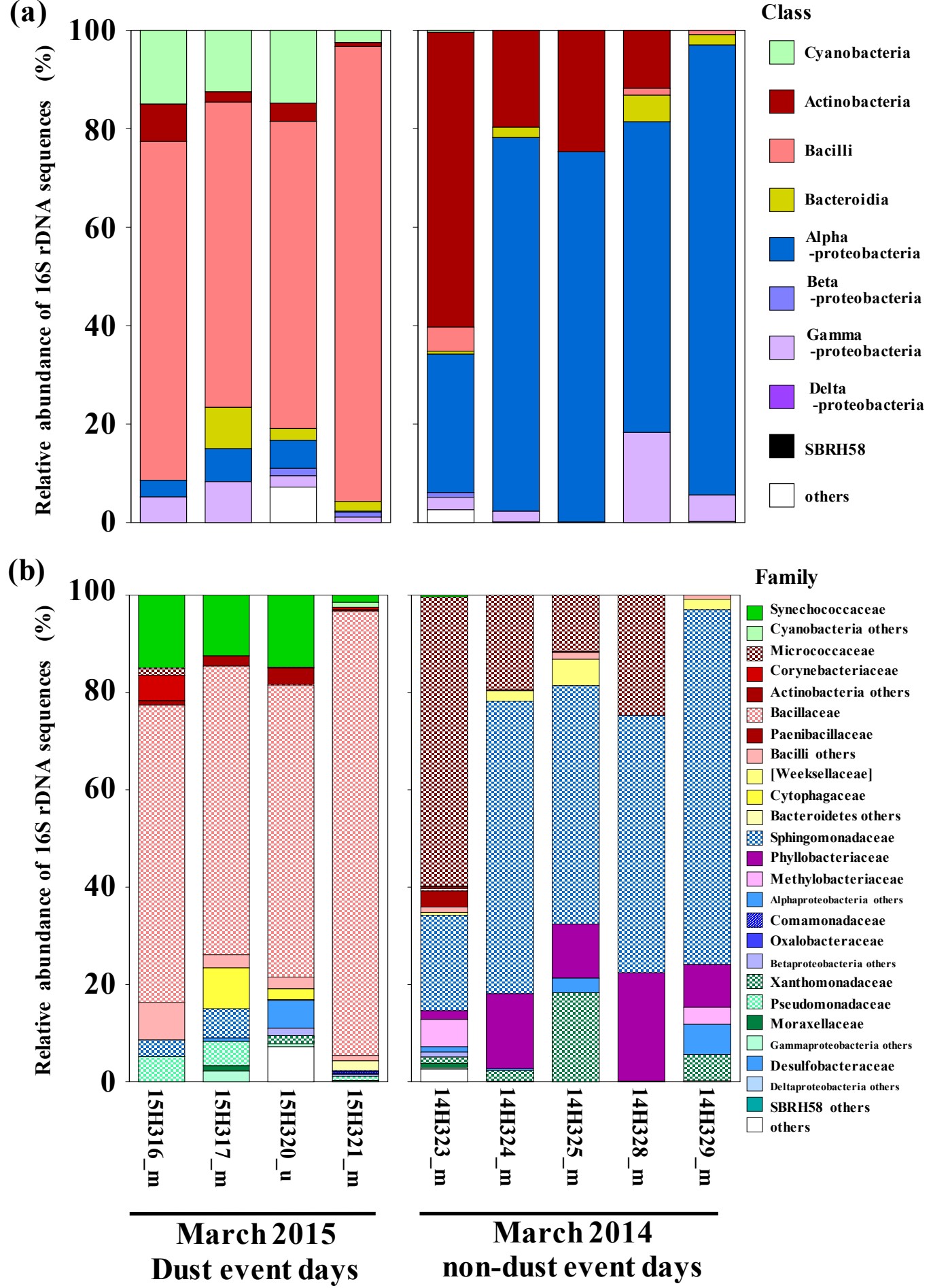

**(a)**

**(b)**

March 2015
Dust event days

March 2014
non-dust event days

Fig. 7 T.Maki et al.

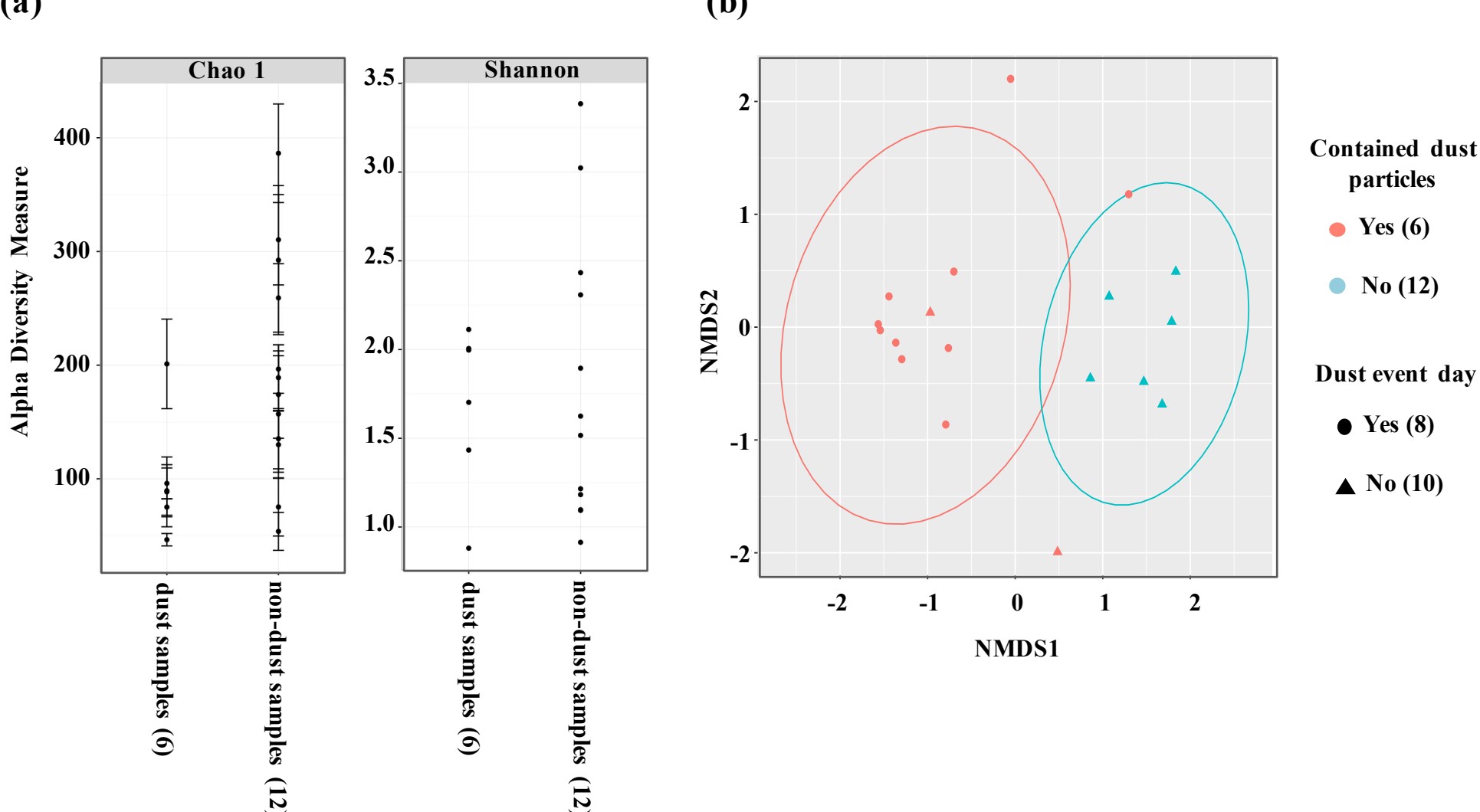

Fig. 8 T. Maki et al.