# Peer review of "Variations in airborne bacterial communities at high altitudes over the Noto Peninsula (Japan) in response to Asian dust events"

_Atmospheric Chemistry and Physics, 2016_

## Referee Comment (RC1) · Anonymous Referee #1 · 14 Jun 2017

In the manuscript entitled "Variations in airborne bacterial communities at high altitudes over the Noto Peninsula (Japan) in response to Asian dust events" by Maki et al., the authors present observation on microbial communities in dust events over Japan, at high medium and low altitudes. A very important topic, and should be in depth explored, yet some modifications and corrections are needed before acceptance for publication.

Specific comments:

- The authors should make it clearer to the readers what is dust and non-dust events. This should be emphasized in the figures (2, 3, 4, 5, 6, 7, 9); figure captions; table (I would recommend adding another column for that information); as well as in the result

text. Otherwise the data presented is somehow confusing and not clear.

- It would be helpful to add some information on the DAPI-staining colors in the introduction part. Introducing these definitions only in the discussion (line 465) makes it hard to follow along the text beforehand.

- line 103: It is specified that aerosol origin is from continental areas, however, trajectories and analysis shows marine contribution as well. please rephrase.

- line 120: How were the filter sterilized? please add either company cat. number, or sterilization technique.

- line 160: Please add the immersion oil type.

- line 174: Reference for the DNA extraction method: Authors should double check the ref., as the Maki 2008 paper refers to the Maki 2004... And - as in the 2004 paper the extraction is not from air filters, the authors should specify the extraction efficiency from filters using this method in the current paper.

- section 3.3: The protease treatment is not detailed in the methodology. Although a very important examination, indicative for protein dominance is yellow particle, no documentation of such treatment and detection before and after treatment is presented. The authors should either supply such results and extend methodology, or remove this part.

- I find it very interesting that marine cyanobacteria contribute to the April 2013, March 2015 events etc. as was also observed by Lang-Yona et al., 2014. This could be relevant for the public health at low altitudes. Please add a discussion on the possible health effects of such species and other gram negative bacteria.

- section 4.2: Organic particles might indeed represent dead bacteria and fungi, however also anthropogenic and natural SOA (especially when air transport over polluted areas, as in the current study). This should be emphasized in the discussion, as the statement (fraction of dead cells compared to total microbes) based on Fig. S4 could

be misleading.

- Line 513: I'm not convinced that cyanobacteria are significantly enriched in dust samples. As described in the result section, cyanobacteria were enriched also in non-dust samples. The authors should supply arguments and statistical evidence for this statement.

-section 4.7: Assuming fluxes of specific bacteria as a representative for the origin of the air mass is a rough estimation and should not be made based on such a study with limited number of sampling points. For example, it is well established that the aerosolization of cyanobacteria would be dominant during bloom events. Therefore, if the authors make such statement of cyanobacteria represent marine-originated aerosols, they should supply evidence for presence of cyanobacteria in high altitudes seasonally and annually, and correlate with bloom events. In addition, one significant source of airborne cyanobacteria are the fresh water bodies. Many other factors affect the abundance of airborne microorganisms, and therefore I find it hard to accept such statement, where the presence of microbes will reflect the origin of the air mass accurately. Authors are requested to restrain their assumption.

-line 671: Please supply reference for this statement.

Technical corrections:

- Section 2.7 should be 2.5.

- line 361-363: Please rewrite this sentence.

- line 421: "…their abundance fluctuated between from…" please check phrasing.

- line 483: .."ranged from 23.3…" – consider rephrasing.

- line 505: Mazar et al. reported dust microbial composition over east Mediterranean areas (not European). Please correct.

- Line 513: Please check if "Figure 4" in the text should be corrected.

- Figure 2 – Caption: should be corrected for black particles denoted in grey color.

- Figure 8b: Authors should better defined symbols. It is not clear (from both legend and caption) what are the blue circles (Are they dust samples? non-dust?) The authors should also add information on the statistics significance of the unifrac test. Consider adding dispersion ellipses with 95% standard deviation confidence interval.

- Figure S4: Please specify in caption/legend what the black and white bars indicate.

---

## Referee Comment (RC2) · Anonymous Referee #2 · 4 Jul 2017

Comment on "Variations in airborne bacterial communities at high altitudes over the Noto Peninsula (Japan) in response to Asian dust events" by T. Maki et al.

Previous studies have shown that biological aerosols probably have a significant impact on environment and climate. Recently bioaerosol-radiation-cloud interaction that is known as important research topic for climate community needs more investigation of bioaerosol spatio-temporal distribution in the atmosphere. The manuscript presents an investigation of atmospheric bioaerosol (bacteria) in Japan, mainly by use of aerosol sampling analysis from aircraft measurements. Then concentration and types of airborne bacteria during dust events and non-dust events could be obtained from fluores-

cent microscopy and 16S rDNA sequencing analysis. The topic is of sufficient interest to the communities of study of atmospheric aerosol (especially bioaerosols), climate as well as human health. In general, I find this manuscript to be of interest for publication and appropriate for ACP. There are several suggestions for improvement listed below that should be considered by the authors and the editors before publication.

1. Introduction: bioaerosols could act as active ice nucleus, consequently affect the microphysical properties of cloud in the atmosphere. Please review some papers about climate effects of bioaerosol, so that the readers are easy to understand the importance of your study.

2. Line 28 in page 3: the authors claimed that aerosols in the two cities directly originate from continental areas. I think it is not rigorous and suitable. There are several sources of aerosols in the Noto Peninsula, such as continental and Ocean area, even from local area, depending on condition of airflows. The word should be changed.

3. Line 23 in page 4: depolarization ratio is more popular for lidar community that depolarization rates. Please replace it throughout the manuscript.

4. Line 8 in page 5: add 'number concentration' to the behind of 'aerosol'.

5. Line 17 in page 6: change 'dust mineral' to 'mineral dust'.

6. Line 7-10 in page 7: the word 'troposphere' is not appropriate in the manuscript, please consider 'tropopause'.

7. Line 25-29 in page 7: please rewrite and cut the paragraph short, it is not necessary to list so many names of the samples. Perhaps the authors can mark dust samples and non-dust samples in Table 1.

8. Section 3.3: four types of fluorescence particles, such as white, blue, yellow, or black particles, could be seen from fluorescent microscopy. To make the reader easier understand, the author should explain the methods and basis of classification. For example, why the white particles are indicative of mineral dust and yellow particles are

organic matter.

9. Section 4.1: I suggest move this sentences to Introduction and Section 3.1. Also, I suggest that rewrite the Section 4, and move some sentences to Introduction.

10. Line 21 in page 12: combine "Maki et al., 2010" and "Maki et al., 2013" to "Maki et al., 2010 and 2013".

11. Line 32 in page 12: add 'long-range' in the front of 'transported'.

12. Figure 1: it is not easy for the readers to understand meaning. Please enlarge four panels of helicopter flight routes and reduce size of the East Asia map. Furthermore, panel (a) can be removed and the location of three cities could be marked in panel (b). N and E should be put at the front of latitude ad longitude, such as 50°N and 120°E.

13. Figure 2: according to the meaning described in the paper, the authors would like to use depolarization ratio of aerosols from lidar measurements, for classifying dust events and non-dust events. But the lidar data as shown in fig. 2 is attenuated backscattering, not depolarization ratio. Same as for the panel (a) in fig. 4 and fig. 5. Please replace the data.

14. In my opinion, more bacteria should be observed during dust events comparing the condition during non-dust events. Because mineral dust usually can be long-range transported with bioaerosols. However, concentration of fluorescent particles (especially blue particles) at near surface (ground level) was lower during dust events (as shown in fig. (a) and (b)) than those duing non-dust events. Please explain the reason.

15. Figure 3: there are several backward trajectories in each panel, but the authors claimed that these three-day backward trajectories only be obtained at two altitudes (2500m and 1200m). Same as for the panel (c) in fig. 4 and fig. 5. Please explain it.

16. Figure 5: the title of x-axis in panel (a) should be "March 2015", please change it.

17. The results in the paper give us more information about bioaerosols in the

atmosphere, especially during dust events. The authors are encouraged to compare their results with others from previous studies. Please summarize similar results from other papers in response to dust events, and then add a table in Section discussion.

Please also note the supplement to this comment: https://www.atmos-chem-phys-discuss.net/acp-2016-1095/acp-2016-1095-RC2-supplement.pdf

---

## Author Comment (AC1) · 8 Aug 2017

Dear Anonymous Referee #1:

We thank for admitting the value of our manuscript very much. I take your comments into account in our revised manuscript. I revised our manuscript with paying attention to the points that you commented. The revised manuscript is attached as supplement file. I described my response for each your comment. The sections [Q] indicate your comments and the sections (A) indicate my responses. The changes introduced in the revised manuscript were indicated by the line numbers at the sections (A).

[Figure]

[Q] The authors should make it clearer to the readers what is dust and non-dust events. This should be emphasized in the figures (2, 3, 4, 5, 6, 7, 9); figure captions; table (I would recommend adding another column for that information); as well as in the result text. Otherwise the data presented is somehow confusing and not clear.

(A) The sampling days of dust or non-dust events have been indicated in Figures and Figure captions in the revised manuscript (Figures 2, 3, 4, 5, 6, and 7). Additional columns defining the dust event days have been inserted into Table 1.

[Q] It would be helpful to add some information on the DAPI-staining colors in the introduction part. Introducing these definitions only in the discussion (line 465) makes it hard to follow along the text beforehand.

(A) Some information on the DAPI-staining colors have been inserted in the Introduction section and the Experiment section in the revised manuscript (lines 89-1091.

[Q] line 103: It is specified that aerosol origin is from continental areas, however, trajectories and analysis shows marine contribution as well. please rephrase.

(A) As this decision, the explanations of aerosol origins over Noto Peninsula were rephrased in the revised manuscript (lines 121-122).

[Q] - line 120: How were the filter sterilized? please add either company cat. number, or sterilization technique.

(A) In the revised manuscript, we have added the information of filter and the filter -sterilization processes (lines 138-142).

[Q] - line 160: Please add the immersion oil type.

(A) The immersion oil type has been inserted in the revised manuscript (lines 181-182).

[Q] - line 174: Reference for the DNA extraction method: Authors should double check the ref., as the Maki 2008 paper refers to the Maki 2004... And - as in the 2004 paper the extraction is not from air filters, the authors should specify the extraction efficiency

from filters using this method in the current paper.

(A) Since gDNA amounts were not enough for the direct determination using light absorbance, the gDNA were determined the PCR products at the first PCR amplification. The extraction efficiency from filters were estimated by the comparison between the PCR products and the particle concentrations by DAPI count, indicating that more 90% of gDNA can be collected by this DNA extraction system. The detail explanations about the DNA extractions have been added to the section of Experiments in the revised manuscript (lines 229-235).

[Q] - section 3.3: The protease treatment is not detailed in the methodology. Although a very important examination, indicative for protein dominance is yellow particle, no documentation of such treatment and detection before and after treatment is presented. The authors should either supply such results and extend methodology, or remove this part.

(A) Although we already have possessed some results about the protease treatments of yellow particles, the data was not sufficient for demonstrating that all yellow particles are composed of protein. Moreover, I think the yellow particle fractions includes unknown organic components. Accordingly, in the revised paper, this part has been removed. The identification of yellow particles are further works.

[Q] - I find it very interesting that marine cyanobacteria contribute to the April 2013, March 2015 events etc. as was also observed by Lang-Yona et al., 2014. This could be relevant for the public health at low altitudes. Please add a discussion on the possible health effects of such species and other gram negative bacteria.

(A) Thank you for your suggestion and the information about valuable reference. We have discussed about the health effects by airborne cyanobacteria with referring to the suggested reference (lines 634-638).

[Q] - section 4.2: Organic particles might indeed represent dead bacteria and fungi,

however also anthropogenic and natural SOA (especially when air transport over polluted areas, as in the current study). This should be emphasized in the discussion, as the statement (fraction of dead cells compared to total microbes) based on Fig. S4 could be misleading.

(A) Thank you for your suggestion. I agree to this comments. The anthropogenic and natural SOA were also included in the yellow fluorescent fractions. This topic has been discussed in the revised manuscript (lines 500-506).

[Q] - Line 513: I'm not convinced that cyanobacteria are significantly enriched in dust samples. As described in the result section, cyanobacteria were enriched also in non dust samples. The authors should supply arguments and statistical evidence for this statement.

(A) In the section of previous manuscript, I mistake to describe about cyanobacteria as the dust specific bacteria. Correctly, cyanobacteria are thought to be the bacterial populations in regardless of dust events and originated from marine environments. The name "cyanobacteria" has been removed at the section of dust-specific bacteria in the revised manuscript (lines 528-529).

[Q] - section 4.7: Assuming fluxes of specific bacteria as a representative for the origin of the air mass is a rough estimation and should not be made based on such a study with limited number of sampling points. For example, it is well established that the aerosolization of cyanobacteria would be dominant during bloom events. Therefore, if the authors make such statement of cyanobacteria represent marine-originated aerosols, they should supply evidence for presence of cyanobacteria in high altitudes seasonally and annually, and correlate with bloom events. In addition, one significant source of airborne cyanobacteria are the fresh water bodies. Many other factors affect the abundance of airborne microorganisms, and therefore I find it hard to accept such statement, where the presence of microbes will reflect the origin of the air mass accurately. Authors are requested to restrain their assumption.

(A) I agree to your comments. We need sufficient information obtained from more numbers of air samples and detail discussion for establishing the air-mass tracking by bacterial compositions. Then this section has been removed and the shortage description about the tracking idea was indicated in the section of Conclusion (lines 659-672).

[Q] - line 671: Please supply reference for this statement.

(A) This parts have been eliminated, because this description about bioaerosol tracking have been shortened and removed to the Conclusion section.

Technical corrections:

[Q] - Section 2.7 should be 2.5.

(A) Section 2.7 has been revised to 2.5 (line 251).

[Q] - line 361-363: Please rewrite this sentence.

(A) I have revised this sentence (lines 378-381).

[Q] - line 421: ": : :their abundance fluctuated between from: : :" please check phrasing.

(A) Sorry for mistake. I have revised this phrase (line 435).

[Q] - line 483: .."ranged from 23.3: : :" – consider rephrasing.

(A) I have rephrased this section in the revised manuscript (lines 495-496).

[Q] - line 505: Mazar et al. reported dust microbial composition over east Mediterranean areas (not European). Please correct.

(A) I'm sorry for errors. " European " has been revised to " east Mediterranean areas " (line 519).

[Q] - Line 513: Please check if "Figure 4" in the text should be corrected.

(A) Sorry for mistake. I have changed to "Figure 4" (line 529).

**[ACPD](ACPD)**

Interactive
comment

[Q] - Figure 2 – Caption: should be corrected for black particles denoted in grey color.

(A) The caption has been revised to indicate the matching color (line 1002).

[Q] - Figure 8b: Authors should better defined symbols. It is not clear (from both legend and caption) what are the blue circles (Are they dust samples? non-dust?) The authors should also add information on the statistics significance of the unifrac test. Consider adding dispersion ellipses with 95% standard deviation confidence interval.

(A) I agree to your comment. The definition for each sample was not clear. After the characteristics of samples have been improved to be defined, Figure 8b and its figure caption has been revised to eliminate the confusion relating to symbols (Figure 8b).

[Q] - Figure S4: Please specify in caption/legend what the black and white bars indicate.

(A) The caption of Figure S4 has been improved in the revised manuscript (Figure S4).

Please also note the supplement to this comment:
https://www.atmos-chem-phys-discuss.net/acp-2016-1095/acp-2016-1095-AC1-supplement.pdf
* * *
[Figure]

Fig. 1 T. Maki et al.

**Fig. 1.** Revised Figure 1

[Figure]

Fig. 2 T. Maki et al.

**Fig. 2.** Revised Figure 2

[Figure]

Fig. 3 T.Maki et al.

**Fig. 3.** Revised Figure 3

[Figure]

Fig. 4 T.Maki et al.

**Fig. 4.** Revised Figure 4

[Figure]

[Figure]

[Figure]

Fig. 5 T.Maki et al.

**Fig. 5.** Revised Figure 5

[Figure]

Fig. 6 T.Maki et al.

**Fig. 6.** Revised Figure 6

[Figure]

Fig. 7 T.Maki et al.

**Fig. 7.** Revised Figure 7

[Figure]

Fig. 8 T. Maki et al.

**Fig. 8.** Revised Figure 8

**Table 1 Sampling information during the sampling periods.**

[revised manuscript text omitted]

Fig. S4. Ratios of yellow fluorescence particles to the total of yellow and microbial particles. (a) The bioaerosol samples were collected at the three or two altitudes over the Noto Peninsula on 19 March 2013 (LT), 28 April 2013 (LT), 28 March 2014 (LT), and 20 March 2015 (LT) and at the altitudes of 1,200 m (except for the 500 m of 20 March 2015) over the Noto Peninsula from 16 to 23 March in 2015 (LT), and from 23 to 29 March in 2014 (LT). Dust samples and non-dust samples were indicated using black bars and white bars, respectively. (b) The average ratios of Dust samples and non-dust samples.

**Fig. 11.** Revised Figure S4

---

## Author Comment (AC2) · 8 Aug 2017

Dear Anonymous Referee #2:

We thank for admitting the value of our manuscript very much. I take your comments into account in our revised manuscript. I revised our manuscript with paying attention to the points that you commented. I described my response for each your comment. The revised manuscript is attached as supplement file. The sections [Q] indicate your comments and the sections (A) indicate my responses. The changes introduced in the revised manuscript were indicated by the line numbers at the sections (A).

[Figure]

[Q]1. Introduction: bioaerosols could act as active ice nucleus, consequently affect the microphysical properties of cloud in the atmosphere. Please review some papers about climate effects of bioaerosol, so that the readers are easy to understand the importance of your study.

(A1) The climate effects of bioaerosol has been enhanced using some references in the Introduction section (lines 45-59).

[Q]2. Line 28 in page 3: the authors claimed that aerosols in the two cities directly originate from continental areas. I think it is not rigorous and suitable. There are several sources of aerosols in the Noto Peninsula, such as continental and Ocean area, even from local area, depending on condition of airflows. The word should be changed.

(A2) I agree with this comment. Several sources areas of air-mass transported to Noto Peninsula were explained in the revised manuscript (lines 121-122).

[Q]3. Line 23 in page 4: depolarization ratio is more popular for lidar community that depolarization rates. Please replace it throughout the manuscript.

(A3) The term "depolarization rates" has been changed to "depolarization ratio" in the revised manuscript (entire revised manuscript).

[Q]4. Line 8 in page 5: add 'number concentration' to the behind of 'aerosol'.

(A4) Thank you for your indication. I have revised this part (lines 195-196).

[Q]5. Line 17 in page 6: change 'dust mineral' to 'mineral dust'.

(A5) As your decision, I have changed the term 'dust mineral' to 'mineral dust' (entire revised manuscript).

[Q]6. Line 7-10 in page 7: the word 'troposphere' is not appropriate in the manuscript, please consider 'tropopause'.

(A6) Thank you for your suggestion. In this section, I have revised to more clear explanation defining the boundary layers over sampling areas (lines 286-288).

[Q]7. Line 25-29 in page 7: please rewrite and cut the paragraph short, it is not necessary to list so many names of the samples. Perhaps the authors can mark dust samples and non-dust samples in Table 1.

(A7) I also think Table 1 can cover the explanation about sample names. Accordingly, this parts explaining about the sample name have been shortened in the revised manuscript (lines 321-325).

[Q]8. Section 3.3: four types of fluorescence particles, such as white, blue, yellow, or black particles, could be seen from fluorescent microscopy. To make the reader easier understand, the author should explain the methods and basis of classification. For example, why the white particles are indicative of mineral dust and yellow particles are organic matter.

(A8) Although some parts of the DAPI staining theory of each fluorescent particles are unclear, they were tried to be explained in the revised manuscript (lines 188-195).

[Q]9. Section 4.1: I suggest move this sentences to Introduction and Section 3.1. Also, I suggest that rewrite the Section 4, and move some sentences to Introduction.

(A9) I agree to your comments. The previous discussion section included some parts which had to be moved to Introduction. In the revised manuscript, the parts were shortened and move to Introduction and the introduction has been modified (in particular lines 455-459, 517-522).

[Q]10. Line 21 in page 12: combine "Maki et al., 2010" and "Maki et al., 2013" to "Maki et al., 2010 and 2013".

(A10) Thank you for your suggestion. "Maki et al., 2010" and "Maki et al., 2013" have been combined to "Maki et al., 2010 and 2013" in the revised manuscript (line 551).

[Q]11. Line 32 in page 12: add 'long-range' in the front of 'transported'.

(A11) The term 'long-range' has been moved to the front of 'transported' (line 567).

[Q]12. Figure 1: it is not easy for the readers to understand meaning. Please enlarge four panels of helicopter flight routes and reduce size of the East Asia map. Furthermore, panel (a) can be removed and the location of three cities could be marked in panel (b). N and E should be put at the front of latitude ad longitude, such as 50°N and 120°E.

(A12) The maps in Figure 1 have been improved by depending on your suggestion. Thank you for your comments (Figure 1).

[Q]13. Figure 2: according to the meaning described in the paper, the authors would like to use depolarization ratio of aerosols from lidar measurements, for classifying dust events and non-dust events. But the lidar data as shown in fig. 2 is attenuated backscattering, not depolarization ratio. Same as for the panel (a) in fig. 4 and fig. 5. Please replace the data.

(A13) In the previous manuscript, the data in Figs. 2, 4 and 5 were originated from depolarization ratio, but I showed wrong scale bar and unit. Sorry for causing confusion. The scale bar and unit have been changed to correct ones in the revised manuscript (Figures. 2, 4 and 5). Furthermore, the explanation about depolarization ratio have been also revised for describing that the ratio means the rates of non-spherical aerosols among all particles (lines 162-164).

[Q]14. In my opinion, more bacteria should be observed during dust events comparing the condition during non-dust events. Because mineral dust usually can be long-range transported with bioaerosols. However, concentration of fluorescent particles (especially blue particles) at near surface (ground level) was lower during dust events (as shown in fig. (a) and (b)) than those during non-dust events. Please explain the reason.

(A14) On our opinion, the fluorescent particles (blue particles and others) are mostly similar each other between fig. (a) and (b), because the particle concentration units of x axis for fig. (a) are one order higher that for fig. (b); fig. (a): 106 particles/m3, and fig. (b): 105 particles/m3. However, I think that the reason for the similar concentrations is needed for this paper and should be inserted in the revised manuscript. At this sampling periods, the high amounts of bioaerosols would be transported to high altitudes and have not fall down to ground surfaces. On the other hands, the air mass during non-dust events is thought to including high amounts of local aerosols. Accordingly, the microbial concentrations in non-dust events were higher than those of dust events. This explanation has been inserted in the revised manuscript (lines 479-484).

[Q]15. Figure 3: there are several backward trajectories in each panel, but the authors claimed that these three-day backward trajectories only be obtained at two altitudes (2500m and 1200m). Same as for the panel (c) in fig. 4 and fig. 5. Please explain it.

(A15) Trajectories at two altitudes (2500m and 1200m) were calculated at every hour for 4hr (0hr, 1hr, 2hr, 3hr and 4hr) before the sampling finish time of each sampling periods. Accordingly, there are total 10 trajectories for each panel. This explanation has been inserted in the captions of Figs. 3, 4 and 5 (lines 1005-1006, 1019-1020, 1033-1034).

[Q]16. Figure 5: the title of x-axis in panel (a) should be "March 2015", please change it.

(A16) Sorry. I have changed "March 2014" to "March 2015" (Figure 5).

[Q]17. The results in the paper give us more information about bioaerosols in the atmosphere, especially during dust events. The authors are encouraged to compare their results with others from previous studies. Please summarize similar results from other papers in response to dust events, and then add a table in Section discussion.

(A17) As your comment, more references have been cited and the bacterial communities differed from the data of previous researches was discussed in the revised manuscript (Sections of Introduction and Discussion, Table 2).

Please also note the supplement to this comment:
https://www.atmos-chem-phys-discuss.net/acp-2016-1095/acp-2016-1095-AC2-
supplement.pdf

[Figure]

[Figure]

Fig. 1 T. Maki et al.

**Fig. 1.** Revised Figure 1

[Figure]

Fig. 2 T. Maki et al.

**Fig. 2.** Revised Figure 2

[Figure]

Fig. 3 T.Maki et al.

**Fig. 3.** Revised Figure 3

[Figure]

Fig. 4 T.Maki et al.

**Fig. 4.** Revised Figure 4

**(a)**

[Figure]

**(b)**

[Figure]

**(c)**

[Figure]

Fig. 5 T.Maki et al.

**Fig. 5.** Revised Figure 5

[Figure]

Fig. 6 T.Maki et al.

**Fig. 6.** Revised Figure 6

[Figure]

Fig. 7 T.Maki et al.

**Fig. 7.** Revised Figure 7

[Figure]

[Figure]

Fig. 8 T. Maki et al.

**Fig. 8.** Revised Figure 8

**Table 1 Sampling information during the sampling periods.**

[revised manuscript text omitted]